# SECA: Self-Guided Model Calibration

## Abstract

Deep learning models frequently exhibit poor calibration, where predicted confidence scores fail to align with actual accuracy rates, undermining model reliability in safety-critical applications. We propose a novel train-time calibration method named **SECA** (**Se**lf-guided Model **Ca**libration), a **hyper-parameter-free** approach designed to improve predictive calibration through dynamic confidence regularisation. SECA constructs adaptive soft targets by fusing batch-averaged model predictions with one-hot ground-truth labels during training, thereby creating a self-adaptive calibration mechanism that adapts target distributions based on the model's predictive behaviour. This leads to well-calibrated predictions without additional hyper-parameter tuning or significant computational overhead. Our theoretical analysis elucidates SECA's underlying mechanisms from entropy regularisation, gradient dynamics, and knowledge distillation perspectives. Extensive empirical evaluation demonstrates that SECA consistently achieves superior calibration performance compared to the Cross-Entropy loss and other state-of-the-art calibration methods across diverse architectures (CNN, ViT, BERT) and benchmark datasets in visual recognition and natural language understanding.[1]

## 1    Introduction

Modern deep neural networks are highly effective in achieving remarkable predictive accuracy across various domains, yet they frequently exhibit poor calibration in their predictions. Poor calibration occurs when a model's predicted confidence scores do not reliably correspond to its actual accuracy rates, e.g., predicting a class with 90% confidence while being correct only 70% of the time (Guo et al., 2017; Minderer et al., 2021). This miscalibration reduces model trustworthiness and may present severe consequences for decision-making, especially in safety-critical real-world scenarios, such as autonomous driving and medical diagnostics (Esteva et al., 2017; Kuutti et al., 2018). The root cause of miscalibration in deep neural networks is primarily due to the optimisation of standard training objectives, such as the Cross-Entropy loss, which aggressively push models to maximise confidence in correct predictions without ensuring that confidence levels align with actual correctness rates (Guo et al., 2017; Wang et al., 2021). Additionally, large model capacities and high-dimensional parameter spaces enable neural networks to over-fit training data, leading to poorly calibrated predictions even on uncertain or ambiguous cases (Müller et al., 2019).

To address this problem, a variety of approaches have been proposed, broadly categorised into post-hoc calibration and train-time calibration methods. Post-hoc methods such as Temperature Scaling (Guo et al., 2017), Isotonic Regression (Zadrozny & Elkan, 2002), and Dirichlet Calibration (Kull et al., 2019a) calibrate model predictions after training via a hold-out validation set. While effective on in-distribution data, these methods require additional data splits and offer no impact on the model's intrinsic calibration behaviour during training, limiting their effectiveness under distribution shift where the training and test distributions differ significantly (Ovadia et al., 2019). In contrast, train-time techniques such as Label Smoothing Müller et al. (2019), Focal Loss (Lin et al., 2017) and Dual Focal Loss (Tao et al., 2023) modify the loss function to improve the model's calibration throughout training. However, these methods typically introduce additional hyper-parameters, e.g., smoothing factor or focusing parameter, that require careful tuning for each dataset or model.

Recent adaptive approaches, such as AdaFocal (Ghosh et al., 2022) and MDCA (Hebbalaguppe et al., 2022), attempt to automatically adjust calibration strength based on prediction dynamics.

---

[1]Source code is available in the **supplementary material**

OLS (Zhang et al., 2021) leverages correct predictions from each training epoch to form an extra loss component. While they improve calibration robustness, these methods often involve validation-based heuristics or auxiliary loss terms, increasing implementation complexity and computational overhead. Furthermore, the majority of these methods have been evaluated primarily on convolutional neural networks (CNNs), with limited exploration of their applicability to modern architectures such as Vision Transformer (ViT) (Dosovitskiy et al., 2021) or BERT (Devlin et al., 2019). These limitations call for a more practical and generalisable approach that is effective across architectures and domains without relying on hyper-parameter tuning or post-training adjustments.

To this end, we propose SECA (Self-Guided Model Calibration), an effective self-guided loss function designed to improve model calibration without introducing extra hyper-parameters. SECA enhances the standard Cross-Entropy loss by incorporating a hybrid label component derived directly from the model's own batch-level predicted probability distribution. Specifically, instead of using a fixed one-hot ground-truth vector as the training target, SECA dynamically constructs a hybrid label that combines the one-hot label with a softened target based on the model's current prediction. This probabilistic soft label acts as a self-guided correction signal that promotes well-calibrated learning by adjusting the certainty of target labels proportionally to the model's prediction behaviour. As training progresses, this mechanism allows the model to self-regulate its confidence levels, encouraging appropriately calibrated predictions when miscalibration arises, without penalising learning when the model exhibits genuine uncertainty. The primary contributions of this work are as follows:

- We introduce SECA for self-guided model calibration, a novel, hyper-parameter-free loss function that dynamically adjusts target distributions using batch-level averaged predictions. Unlike conventional calibration techniques, SECA requires no auxiliary loss terms or manual tuning, significantly simplifying its practical adoption across diverse domains and architectures.

- We provide a rigorous theoretical analysis along with empirical studies, from entropy regularisation, gradient dynamics, and knowledge distillation perspectives to reveal how SECA effectively improves neural network calibration by leveraging the collective behaviour of batch samples, thus promoting well-calibrated training.

- We empirically validate SECA through extensive experiments across a wide range of benchmark datasets, including visual recognition (CIFAR-10/100, ImageNet) and natural language understanding (DBpedia, 20 Newsgroups), using representative architectures such as ResNet, ViT, and BERT. Our results demonstrate that SECA consistently improves calibration, achieving substantial reductions in Static Calibration Error (SCE), Expected Calibration Error (ECE), and Adaptive ECE (AECE) compared with other SoTA methods.

## 2 RELATED WORK

Model calibration, where predicted probabilities should reliably correspond to true correctness likelihoods, has long been recognised as a critical issue in machine learning. Before the deep learning era, calibration was studied in the context of classical models such as logistic regression, support vector machines, and decision trees (Brier, 1950; Platt et al., 1999b; Niculescu-Mizil & Caruana, 2005; Zadrozny & Elkan, 2001b). Guo et al. (2017) later demonstrated that modern deep learning architectures such as ResNets and Inception often produce poorly calibrated predictions, particularly when trained with the standard Cross-Entropy loss. To measure calibration quality, they adopted the Expected Calibration Error (ECE), which has since become a standard metric in calibration research. Given the widespread recognition of calibration issues in deep learning, numerous approaches have been developed to address this problem. Broadly, calibration methods fall into two categories: post-hoc calibration and train-time calibration.

**Post-Hoc Calibration.** Post-hoc methods calibrate models after training, without modifying the learned parameters (Naeini et al., 2015; Guo et al., 2017; Kull et al., 2017; 2019b; Wenger et al., 2020; Ding et al., 2021). For instance, Guo et al. (2017) introduced Temperature Scaling, a simple yet effective technique that scales logits uniformly using a single temperature parameter, tuned on a validation set. Other post-hoc methods include histogram binning (Zadrozny & Elkan, 2001a), Platt scaling (Platt et al., 1999a), Isotonic Regression (Zadrozny & Elkan, 2002) and Dirichlet calibration (Kull et al., 2019a), which offer varying trade-offs between complexity and flexibility. While effective, post-hoc methods do not improve the model's intrinsic uncertainty or robustness under

distribution shift (Minderer et al., 2021; Ovadia et al., 2019). Thus, this study will primarily focus on train-time calibration techniques.

**Train-Time Calibration.** In contrast to post-hoc calibration techniques that adjust a model's outputs after training, train-time calibration methods aim to directly shape the model's confidence behaviour during the training process. These techniques typically modify the loss function or training targets to encourage well-calibrated predictions and promote better-aligned probabilistic outputs.

One of the most widely adopted techniques is label smoothing, which was initially proposed by Szegedy et al. (2016) and systematically studied for calibration by Müller et al. (2019). Label smoothing replaces one-hot encoded targets with softened label distributions, which not only acts as a form of regularisation but also leads to improved calibration, as it encourages the model to produce well-calibrated probability estimates rather than over-confident predictions. However, the optimal smoothing factor $\alpha$ requires careful tuning across models and datasets. Another prominent technique is the focal loss (Lin et al., 2017), which initially aimed to address class imbalance in object detection. It dynamically down-weights the contribution of well-classified/high-confidence samples using a focusing parameter $\gamma$, thus forcing the model to focus on harder examples. Mukhoti et al. (2020) later observed that, with appropriate choice of $\gamma$, focal loss also yields better calibrated predictions, particularly in imbalanced and long-tailed scenarios. Building upon this foundation, several extensions have been developed to enhance focal loss's calibration capabilities. AdaFocal (Ghosh et al., 2022) builds upon focal loss by learning a sample group-specific focusing parameter $\gamma$ using gradient-based meta-learning, allowing the calibration strength to adapt dynamically based on the model's behaviour during training. Whilst it demonstrates improved calibration across various networks and datasets, the approach necessitates a validation set for meta-updating. Dual Focal Loss (DFL) (Tao et al., 2023) extends focal loss by simultaneously considering both the confidence of the correct class and that of the most competitive incorrect class, addressing calibration imbalances more effectively than standard focal loss through explicit modelling of this relationship. A recent method, MDCA, introduces a calibration-specific regularisation term that aligns the model's class-wise confidence with empirical class-wise accuracy (Hebbalaguppe et al., 2022), helping to control per-class miscalibration. MDCA shows promising improvements in both Static Calibration Error (SCE) and Expected Calibration Error (ECE) metrics.

While train-time calibration methods have proven effective, most of them require manual tuning or complex adaptation mechanisms to achieve well-calibrated predictions across classes and difficulty levels. In contrast, our proposed SECA is hyper-parameter-free and automatically adapts per-class and per-batch based on the model's own predictive behaviour. It achieves strong calibration performance without the need for auxiliary components or validation-based tuning, offering a practical and generalisable alternative to existing train-time methods.

## 3 METHODOLOGY

In this section, we first revisit the root cause of the over-confidence issue. Subsequently, we introduce SECA, a novel loss function for self-guided model calibration. Finally, we provide theoretical analysis regarding the mechanisms of SECA, from entropy regularisation, gradient dynamics, and knowledge distillation perspectives, respectively.

### 3.1 PRELIMINARIES

In a standard supervised classification setup, let $\mathcal{D} = \{(\mathbf{x}_i, y_i)\}_{i=1}^{N}$ be the training dataset, where $\mathbf{x}_i \in \mathbb{R}^d$ is the input and $y_i \in \{1, \ldots, C\}$ is the corresponding ground-truth class label of $C$ possible classes. Let $f_\theta : \mathbb{R}^d \to \mathbb{R}^C$ be a neural network parametrised by $\theta$, producing a logit vector $\mathbf{z}_i = f_\theta(\mathbf{x}_i) \in \mathbb{R}^C$. The predicted probability vector for the $i$-th sample is given by the *softmax*:

$$\mathbf{p}_i = \text{softmax}(\mathbf{z}_i) = [p_{i,1}, p_{i,2}, \ldots, p_{i,C}]^\top, \quad \text{where} \quad p_{i,c} = \frac{\exp(z_{i,c})}{\sum_{j=1}^{C} \exp(z_{i,j})}, \quad (1)$$

is the probability for class $c$, given the logits $\mathbf{z}_i$. The model prediction is calibrated if for all confidence levels $p \in [0, 1]$ Guo et al. (2017), the following holds:

$$\mathbb{P}\left(Y = \hat{Y} \mid \max_c P_c = p\right) = p, \quad (2)$$

where $\hat{Y}$ denotes the predicted label, $\max_c P_c$ is the model's maximum confidence, and $\mathbb{P}$ represents the empirical probability that the model is correct given that it is predicting with confidence $p$. In other words, if the model predicts a class with 80% confidence, it should be correct approximately 80% of the time for those predictions. However, modern neural networks often violate this condition, with miscalibration becoming increasingly severe as predicted confidence increases.

**Cross-Entropy.** The Cross-Entropy loss is the most commonly used objective for classification:

$$\mathcal{L}_{\mathrm{CE}} = -\log p_{i,y_i}, \tag{3}$$

where $p_{i,y_i}$ is the predicted probability for the correct class $y_i$. It assumes a one-hot target vector $\mathbf{q}_i \in \{0,1\}^C$ with $\sum_{c=1}^{C} q_{i,c} = 1$, where $q_{i,c} = \mathbb{I}[c = y_i]$ and $\mathbb{I}[\cdot]$ is the indicator function. The gradient with respect to the correct class logit is $\frac{\partial \mathcal{L}_{\mathrm{CE}}}{\partial z_{i,y_i}} = p_{i,y_i} - 1$, which remains negative even when $p_{i,y_i}$ approaches 1, continuously driving the logit upwards. This persistent gradient pressure reinforces already confident predictions and is a primary cause of overconfidence (Guo et al., 2017).

**Note on Calibration Scope.** While neural networks can exhibit both over-confidence and under-confidence, modern deep networks trained with Cross-Entropy loss predominantly suffer from over-confidence, particularly after convergence (Guo et al., 2017; Minderer et al., 2021). Under-confidence typically occurs in early training stages or under severe regularisation, but is less prevalent in standard training regimes. Therefore, we focus on over-confidence as the primary calibration challenge, noting that our proposed SECA naturally adapts to both scenarios through its batch-aware mechanism, as we demonstrate in our theoretical analysis (Section 3.3).

## 3.2 SELF-GUIDED MODEL CALIBRATION VIA SECA

To calibrate the model during training, we propose SECA, an intuitive yet effective loss function that dynamically adjusts the target distribution per-class by leveraging the model's own batch-level predictive confidence in a self-guided manner. Unlike label smoothing, which applies a fixed perturbation to the target labels, SECA adaptively constructs soft labels without requiring any additional hyper-parameters.

Given an input sample $\mathbf{x}_i$, the model outputs logits vector $\mathbf{z}_i = f_\theta(\mathbf{x}_i) \in \mathbb{R}^C$. The predicted probability distribution $\mathbf{p}_i$ is computed via the *softmax* function (Eq. 1). For class $j \in \{1,\ldots,C\}$, we define the set of batch samples as $S$, samples belonging to class $j$ as:

$$S_j = \{i \in \{1,\ldots,M\} \mid y_i = j\}, \tag{4}$$

where $M$ is the batch size and $y_i$ denotes the ground-truth label for sample $i$.

**Batch-Level Class-wise Distribution.** For each class $j \in \{1,\ldots,C\}$, we compute the batch-level average predicted probability distribution $\boldsymbol{\mu}_j$ across all samples whose ground-truth label is $j$:

$$\boldsymbol{\mu}_j = \frac{1}{|S_j|} \sum_{i \in S_j} \mathbf{p}_i, \tag{5}$$

where $\boldsymbol{\mu}_j = [\mu_{j,1}, \mu_{j,2}, \ldots, \mu_{j,C}]^\top \in \mathbb{R}^C$ is the average probability distribution for class $j$. Each element $\mu_{j,c}$ represents the average predicted probability for class $c$ among all samples whose true label is $j$. The distribution $\boldsymbol{\mu}_j$ captures the model's collective belief over all classes, conditioned on samples belonging to class $j$ within the current batch.

**Construction of Hybrid Target.** For each sample $i$, we define the hybrid target $\tilde{\mathbf{q}}_i$ by combining the one-hot ground-truth label, $\mathbf{q}_i$, with the batch-averaged prediction for its corresponding class:

$$\tilde{q}_{i,c} = q_{i,c} + \mu_{y_i,c}, \text{ where } q_{i,c} = \begin{cases} 1, \text{ if } c = y_i, \\ 0, \text{otherwise}. \end{cases} \tag{6}$$

The above formulation anchors the soft target at the ground-truth class while adaptively softening it based on the model's collective predictive distribution for that class within the batch. Note that, the hybrid target $\tilde{\mathbf{q}}_i$ is intentionally unnormalised. This preserves the full influence of both the one-hot label and the class-conditional batch-averaged prediction. Cross-Entropy with unnormalised non-negative targets is mathematically valid, and the resulting gradient naturally matches the refined expression in Eq. 12.

**Loss Computation.** The SECA loss for a given sample $i$ is then computed as the Cross-Entropy between the model's predicted probability distribution $\mathbf{p}_i$ and the constructed hybrid target $\tilde{\mathbf{q}}_i$:

$$\mathcal{L}_{\text{SECA}} = -\sum_{c=1}^{C} \tilde{q}_{i,c} \log p_{i,c}. \tag{7}$$

## 3.3 Theoretical Analysis

The design of SECA inherently improves model calibration by introducing a self-guided regularisation mechanism during training. Specifically, SECA encourages the model's output distribution to align not only with the ground-truth label but also with the average prediction behaviour of samples from the same class. In the following parts, we formalise this intuition by showing that the SECA can be interpreted as a modified Cross-Entropy objective augmented with a KL-divergence-like regulariser. We further analyse the per-sample gradients induced by this formulation and explain how they naturally provide bidirectional calibration by moderating excessive confidence while strengthening insufficient confidence, thereby leading to well-calibrated outputs throughout training. Additionally, we provide an interpretation from the knowledge distillation perspective, where batch-level statistics serve as adaptive teachers for calibration.

**Entropy Perspective.** Recall the hybrid target $\tilde{\mathbf{q}}_i$ (Eq. 6), we can decompose SECA (Eq. 7) into:

$$\mathcal{L}_{\text{SECA}} = \underbrace{-\log p_{i,y_i}}_{\text{standard CE}} + \underbrace{\left(-\sum_{c=1}^{C} \mu_{y_i,c} \log p_{i,c}\right)}_{\text{KL-like regulariser}}. \tag{8}$$

The first term is the standard Cross-Entropy loss, which encourages the model to increase the predicted probability $p_{i,y_i}$ for the ground-truth class $y_i$ as much as possible. Minimising this term alone typically drives the network towards poorly calibrated outputs, either over-confident or under-confident depending on the training dynamics. The second term serves as an adaptive calibration regulariser that computes the Cross-Entropy between the batch-level averaged class distribution $\boldsymbol{\mu}_{y_i}$ and the model's own prediction $\mathbf{p}_i$. This regulariser dynamically adapts the target confidence based on the collective behaviour of same-class samples, promoting well-calibrated predictions. The term is equivalent to the Kullback-Leibler (KL) divergence (Kullback & Leibler, 1951) as below:

$$KL(\boldsymbol{\mu}_{y_i} \| \mathbf{p}_i) = -\sum_{c=1}^{C} \mu_{y_i,c} \log\left(\frac{\mu_{y_i,c}}{p_{i,c}}\right) = -H(\boldsymbol{\mu}_{y_i}) - \sum_{c=1}^{C} \mu_{y_i,c} \log p_{i,c}, \tag{9}$$

where $H(\boldsymbol{\mu}_{y_i})$ denotes the entropy of $\boldsymbol{\mu}_{y_i}$. If we rearrange the formula, we obtain:

$$-\sum_{c=1}^{C} \mu_{y_i,c} \log p_{i,c} = KL(\boldsymbol{\mu}_{y_i} \| \mathbf{p}_i) + H(\boldsymbol{\mu}_{y_i}), \tag{10}$$

as $-\sum_{c=1}^{C} \mu_{y_i,c} \log p_{i,c}$ is the KL-like term in Eq. 8. Thus, the overall loss can be equivalently written as:

$$\mathcal{L}_{\text{SECA}} = -\log p_{i,y_i} + KL(\boldsymbol{\mu}_{y_i} \| \mathbf{p}_i) + H(\boldsymbol{\mu}_{y_i}). \tag{11}$$

This decomposition highlights the dual effect of SECA: while the standard Cross-Entropy term promotes correct class prediction, the KL divergence regularisation encourages predictions to align with the batch-informed distribution $\boldsymbol{\mu}_{y_i}$, which serves as an adaptive calibration signal. Unlike fixed regularisers (e.g., label smoothing with constant $\alpha$), $\boldsymbol{\mu}_{y_i}$ is adapted based on current model behaviour, providing entropy-increasing regularisation when predictions are over-confident, and concentrating guidance when predictions are under-confident or scattered. This adaptive mechanism naturally leads to well-calibrated predictions without requiring explicit entropy terms or additional hyper-parameters (empirical study is in Appendix A).

**Gradient Perspective.** To understand the impact of SECA on training dynamics, we also examine the gradients with respect to the model logits. Taking the derivative of per-sample SECA $\mathcal{L}_i$ with respect to the logit $z_{i,c}$ for class $c$, we obtain:

$$\frac{\partial \mathcal{L}_i}{\partial z_{i,c}} = 2p_{i,c} - (q_{i,c} + \mu_{y_i,c}), \tag{12}$$

which follows from the general soft-target cross-entropy derivative $\partial L / \partial z = (\sum_j t_j)p - t$ when the hybrid target $\tilde{q}_i = q_i + \mu_{y_i}$ is unnormalised and satisfies $\sum_c \tilde{q}_{i,c} = 2$.

For the ground-truth class $c = y_i$, the gradient simplifies to:

$$\frac{\partial \mathcal{L}_i}{\partial z_{i,c}} = 2p_{i,c} - (1 + \mu_{y_i,c}). \tag{13}$$

If the model is already confident, i.e., $p_{i,c}$ is large, the gradient becomes weakly negative. However, the additional term, $-(1 + \mu_{y_i,c})$ provides a stronger downward correction than in the normalised case, which remains present even when the model is highly confident. This applies a downward force on the logit, which naturally counteracts the tendency of Cross-Entropy loss to endlessly push the logit upward, effectively preventing extreme overconfidence.

And for non-target classes where $k \neq y_i$, the gradient is given by:

$$\frac{\partial \mathcal{L}_i}{\partial z_{i,k}} = 2p_{i,k} - \mu_{y_i,k}. \tag{14}$$

In this case, if the predicted probability $p_{i,k}$ for a non-target class exceeds the batch-averaged value $\mu_{y_i,k}$, the gradient is positive, thereby pushing the corresponding logits downward and reducing the misplaced confidence. Conversely, if $p_{i,k}$ is too low compared to $\mu_{y_i,k}$, the gradient becomes negative, softly encouraging a slight increase in probability for relevant secondary classes.

Overall, this gradient structure explicitly encourages the model's per-sample prediction $\mathbf{p}_i$ to align with the batch-informed calibration target $\boldsymbol{\mu}_{y_i}$. The factor of 2 amplifies the corrective influence of the hybrid target, preserving the full effect of both $q_i$ and $\mu_{y_i}$ rather than reducing the mechanism to a normalised label-smoothing variant. The gradients provide bidirectional calibration correction: reducing excessive confidence when predictions are too sharp, while strengthening confidence when predictions are too diffuse or misaligned with class-typical patterns. This dynamic adjustment process operates without requiring external supervision, hyper-parameter tuning, or hand-designed entropy penalties (empirical study is in Appendix A).

**Knowledge Distillation Perspective.** The mechanism underlying SECA can also be viewed as a form of adaptive self-distillation (Furlanello et al., 2018; Hinton et al., 2015). Specifically, for each target class $y_i$, the batch-level average prediction distribution $\boldsymbol{\mu}_{y_i}$ serves as an adaptive soft teacher constructed from the model's own outputs over samples belonging to class $y_i$. During training, the network is encouraged to align each individual prediction $\mathbf{p}_i$ not only with the one-hot target but also with the class-informed distribution $\boldsymbol{\mu}_{y_i}$, effectively teaching itself appropriate confidence levels based on the collective prediction behaviour of similar samples. This creates a dynamic calibration mechanism where the "teacher" signal adapts to the current state of model predictions for each class.

As training progresses, this alignment process promotes class-wise calibration consistency. Ideally, the per-sample output $\mathbf{p}_i$ converges towards the batch-averaged prediction over samples that share the same label. At convergence, we can expect:

$$\mathbf{p}_i = \frac{1}{|S_{y_i}|} \sum_{j \in S_{y_i}} \mathbf{p}_j = \boldsymbol{\mu}_{y_i}, \tag{15}$$

where $S_{y_i}$ denotes the set of samples with ground-truth label $y_i$ in the batch. This condition implies that each sample's predictive distribution becomes consistent with the calibrated predictive behaviour of its class, leading to well-calibrated outputs that reflect appropriate uncertainty levels rather than miscalibrated *softmax* distributions (empirical study is in Appendix A).

## 4 EXPERIMENTS

**Network Architectures.** To validate the effectiveness and generality of the proposed SECA, we conduct comprehensive experiments across diverse network architectures, including CNN, ViT, and BERT models. All models are trained from scratch under identical training conditions. Our architectural choices are designed to provide comprehensive evaluation across both established and modern paradigms while maintaining computational feasibility. For CIFAR-10/100 experiments

(Krizhevsky et al., 2009), we adopt ResNet32/56 models following the experimental protocol of MDCA (Hebbalaguppe et al., 2022). For large-scale ImageNet evaluation (Deng et al., 2009), we employ ViT-small, which has become increasingly prevalent for large-scale vision tasks. For natural language understanding tasks, DBpedia (Lang, 1995) and 20 Newsgroups (Zhang et al., 2015), we evaluate on both BERT-small and BERT-base architectures to assess scalability across different model sizes within the Transformer family. This architectural diversity spanning traditional CNNs on CIFAR datasets, modern Vision Transformers on ImageNet, and BERT models for NLP tasks, enables comprehensive evaluation across different inductive biases, model types, and task domains.[2]

**Compared Baselines.** We compare our method against Cross-Entropy loss, Focal loss (Focal) (Lin et al., 2017), Label Smoothing (LS) (Müller et al., 2019), MMCE (Kumar et al., 2018), DCA (Liang et al., 2020), FLSD (Mukhoti et al., 2020), MDCA (Hebbalaguppe et al., 2022), Brier (Brier, 1950), OLS (Zhang et al., 2021), Dual Focal (Tao et al., 2023), and AdaFocal (Ghosh et al., 2022).

**Hyper-Parameters Setups.** For method-specific hyper-parameters, e.g. $\gamma$ and $\beta$ in MDCA (Hebbalaguppe et al., 2022), we adopt the configurations from original papers based on their best reported results. All methods are trained under the same dataset-model pairs and general training hyper-parameters as shown in Table 4 (Appendix B), ensuring a consistent evaluation environment.

**Evaluation Metrics.** We evaluate each method via four metrics: Test Error rate (TE), Static Calibration Error (SCE) (Nixon et al., 2019), Expected Calibration Error (ECE) (Guo et al., 2017), and Adaptive ECE (AECE) (Ding et al., 2020). Details about those metrics are in the Appendix C.

## 4.1 EXPERIMENTS ON VISUAL RECOGNITION TASKS

Table 1 presents the experimental results on CIFAR-10 and CIFAR-100 using ResNet32 and ResNet56 networks. Our proposed SECA demonstrates strong calibration performance across both datasets, with distinct advantages emerging on each evaluation setting. On CIFAR-10, while some specialised methods such as MDCA achieve lower calibration errors in certain metrics, SECA maintains competitive performance across all measures while being hyperparameter-free. SECA consistently outperforms the Cross-Entropy baseline, demonstrating meaningful calibration improvements without sacrificing predictive accuracy. Notably, SECA achieves the lowest test error rates (7.07% and 6.47% for ResNet32 and ResNet56, respectively) amongst all evaluated methods, indicating that the calibration improvements do not come at the expense of model accuracy.

| Methods | CIFAR-10 | | | | | | | | CIFAR-100 | | | | | | | |
| --- | --- | --- | --- | --- | --- | --- | --- | --- | --- | --- | --- | --- | --- | --- | --- | --- |
| | ResNet32 | | | | ResNet56 | | | | ResNet32 | | | | ResNet56 | | | |
| | TE | SCE | ECE | AECE | TE | SCE | ECE | AECE | TE | SCE | ECE | AECE | TE | SCE | ECE | AECE |
| Focal ($\gamma$=3.0) | 7.99 | 10.0 | 4.54 | 4.46 | 7.59 | 9.73 | 4.46 | 4.41 | 31.45 | 2.07 | 2.26 | 2.17 | 28.92 | 2.01 | 2.07 | 1.85 |
| LS ($\alpha$=0.1) | 7.42 | 15.0 | 6.37 | 6.28 | 6.66 | 13.8 | 5.49 | 5.30 | 29.95 | 2.28 | 2.80 | 2.86 | 27.21 | 2.18 | 2.35 | 2.51 |
| MMCE ($\beta$=4.0) | 8.43 | 8.47 | 3.40 | 3.47 | 8.18 | 8.45 | 3.30 | 3.34 | 31.68 | 2.50 | 7.53 | 7.52 | 29.63 | 2.36 | 6.93 | 6.90 |
| DCA ($\beta$=1.0) | 7.53 | 9.02 | 4.24 | 4.22 | 6.93 | 7.37 | 3.34 | 3.29 | 30.03 | 3.15 | 12.0 | 12.0 | 27.48 | 2.95 | 11.1 | 11.1 |
| FLSD ($\gamma$=3.0) | 7.90 | 9.88 | 4.44 | 4.41 | 7.51 | 10.3 | 4.75 | 4.71 | 32.02 | 2.10 | 2.16 | 2.12 | 28.95 | 2.02 | 2.30 | 2.28 |
| MDCA ($\gamma$, $\beta$=1.0) | 7.40 | **5.02** | 1.84 | **1.76** | 7.00 | **4.29** | **1.25** | **1.16** | 30.96 | 2.30 | 5.61 | 5.61 | 28.00 | 2.17 | 5.24 | 5.21 |
| Brier | 7.72 | 6.18 | 2.61 | 2.59 | 7.76 | 5.39 | 2.15 | 2.05 | 33.84 | 2.30 | 5.56 | 5.53 | 30.97 | 2.11 | 4.94 | 4.88 |
| OLS ($\alpha$=0.5) | 7.46 | 7.27 | 3.31 | 3.30 | 7.34 | 6.27 | 2.80 | 2.75 | 30.44 | 2.12 | 4.51 | 4.56 | 27.95 | 1.88 | 2.44 | 2.41 |
| DualFocal ($\gamma$=5.0) | 8.01 | 5.03 | **1.82** | 1.79 | 7.62 | 5.62 | 2.61 | 2.51 | 31.54 | 2.17 | 3.30 | 3.26 | 28.21 | 2.05 | 1.94 | 1.96 |
| AdaFocal | 7.56 | 6.49 | 2.69 | 2.62 | 6.79 | 4.56 | 1.44 | 1.37 | 31.27 | 2.74 | 3.42 | 3.43 | 27.89 | 2.54 | 2.75 | 2.76 |
| CE (baseline) | 7.14 | 8.47 | 3.86 | 3.85 | 6.85 | 6.89 | 3.10 | 3.09 | 30.36 | 2.83 | 10.0 | 10.2 | 27.15 | 2.59 | 9.09 | 9.09 |
| **SECA (Ours)** | **7.07** | 6.67 | 2.94 | 2.84 | **6.47** | 5.75 | 2.41 | 2.44 | **29.82** | **1.89** | **1.90** | 1.95 | **26.97** | **1.79** | **1.71** | **1.74** |

Table 1: Comparison between our method and other methods regarding calibration metrics: Test Error (%), SCE (‰), ECE (%) and AECE (%), for ResNet32/56 on CIFAR10/100 datasets. A lower error is better. Results are averaged values based on five independent trainings.

SECA's advantages become even more pronounced on the more challenging CIFAR-100 dataset, where it consistently achieves the best results across all metrics. On CIFAR-100, SECA obtains the lowest test error rates (29.82% and 26.97% for ResNet32 and ResNet56, respectively) while simultaneously achieving superior calibration with the lowest SCE (1.89‰ and 1.79‰), ECE (1.90% and 1.71%), and AECE (1.95% and 1.74%) values compared to all baseline methods. The substantial calibration improvements are particularly evident when comparing to the baseline, with ECE reduced from 10.0% to 1.90% on CIFAR-100 ResNet32. Compared to recent strong baselines

---

[2]ViT-ImageNet experiments are conducted with eight Nvidia RTX 4090 GPUs, and the other experiments are performed using two RTX A5500 GPUs.

including DualFocal and AdaFocal, SECA shows particularly strong performance on this multi-class scenario where the larger number of classes and finer-grained distinctions make calibration more challenging. This demonstrates the effectiveness of SECA's adaptive, class-wise regularisation mechanism in scenarios with higher class complexity.

Table 2 presents the Top-1/-5 test errors and calibration metrics for ViT-small on the ImageNet dataset. SECA demonstrates excellent calibration performance, achieving the best results across all calibration metrics with SCE (0.53‰), ECE (7.47%), and AECE (7.65%), substantially outperforming the Cross-Entropy baseline and other calibration-focused methods. Notably, SECA reduces ECE by 37.3% compared to the Cross-Entropy baseline (from 11.9 to 7.47%) while simultaneously improving Top-1 accuracy from 25.61% to 23.94%.

| Methods | ViT-small on ImageNet | | | | | Training Cost |
|---|---|---|---|---|---|---|
| | TE (Top-1) | TE (Top-5) | SCE | ECE | AECE | |
| Focal ($\gamma$=3.0) | 27.17 | 8.17 | 0.66 | 8.12 | 7.86 | 40.23 Hours |
| LS ($\alpha$=0.1) | 23.78 | 7.80 | 0.68 | 8.99 | 8.82 | 33.85 Hours |
| MMCE ($\beta$=2.0) | 25.61 | 8.64 | 0.58 | 11.8 | 10.9 | 40.13 Hours |
| DCA ($\beta$=1.0) | 25.45 | 8.68 | 0.55 | 12.1 | 11.3 | 40.32 Hours |
| FLSD ($\gamma$=3.0) | 26.12 | 8.31 | 0.66 | 8.25 | 8.05 | 42.85 Hours |
| MDCA ($\gamma$, $\beta$=1.0) | 25.85 | 8.49 | 0.59 | 9.29 | 8.61 | 43.60 Hours |
| Brier | 25.77 | 9.70 | 0.59 | 9.39 | 8.68 | 44.27 Hours |
| OLS ($\alpha$=0.5) | **22.84** | **7.21** | 0.64 | 9.05 | 8.68 | 37.53 Hours |
| DualFocal ($\gamma$=5.0) | 25.80 | 8.47 | 0.65 | 8.23 | 7.80 | 40.69 Hours |
| AdaFocal | 24.74 | 8.31 | 0.61 | 8.31 | 7.67 | 41.31 Hours |
| CE (baseline) | 25.61 | 8.73 | 0.56 | 11.9 | 11.0 | **30.83 Hours** |
| **SECA (Ours)** | 23.94 | 8.03 | **0.53** | **7.47** | **7.65** | 32.08 Hours |

Table 2: Comparison between our method and other methods regarding calibration metrics: Test Error (%), SCE (‰), ECE (%) and AECE (%), for ViT-small on the ImageNet dataset. A lower error or training cost is better.

While OLS achieves the lowest Top-1 and Top-5 error rates (22.84% and 7.21%, respectively), SECA strikes an excellent balance between accuracy and calibration, obtaining competitive accuracy (23.94% Top-1, 8.03% Top-5) while significantly surpassing OLS in calibration quality (ECE: 7.47% vs 9.05%). Among methods that achieve similar accuracy levels, SECA provides substantially better calibration—for instance, compared to Label Smoothing which achieves 23.78% Top-1 error, SECA reduces ECE from 8.99% to 7.47%.

Furthermore, SECA demonstrates excellent computational efficiency with a training cost of 32.08 hours, remaining close to the baseline Cross-Entropy (30.83 hours) while being considerably faster than more complex methods such as MDCA (43.60 hours). This efficiency advantage, combined with its hyper-parameter-free nature, makes SECA particularly attractive for large-scale applications like ImageNet where computational resources are a significant consideration.

To further illustrate the calibration improvements, Figure 1 presents reliability diagrams comparing SECA against several baseline methods on CIFAR-100 with ResNet-56. The diagrams demonstrate that whilst Cross Entropy exhibits substantial overconfidence (particularly in high-confidence bins), SECA maintains close alignment between predicted confidence and actual accuracy across all confidence ranges. More reliability analysis is in Appendix F.

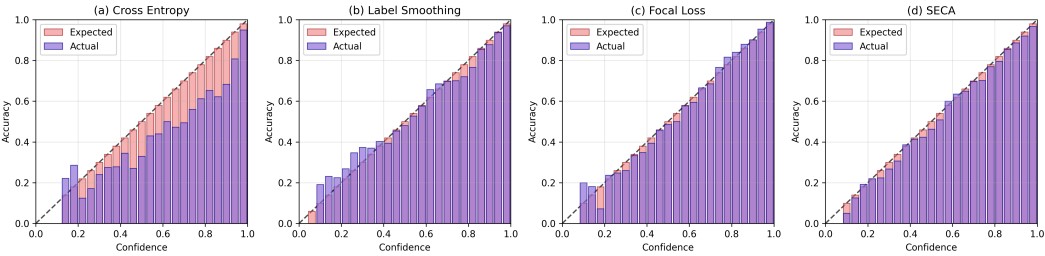

Figure 1: Reliability Diagram of Different Methods on CIFAR-100 with ResNet56.

## 4.2 Experiments on Natural Language Understanding Tasks

Table 3 presents the results on natural language understanding datasets DBpedia and 20 News-groups using BERT-small and BERT-base models. SECA demonstrates strong performance across both datasets, with particularly notable results varying by dataset characteristics. On the DBpedia dataset, SECA achieves excellent overall performance, obtaining the lowest test error rates (1.18% for BERT-small and 1.38% for BERT-base) while maintaining competitive calibration. For ECE and AECE metrics, SECA consistently achieves the best performance across both model sizes (e.g., ECE of 0.35% vs baseline's 0.53% for BERT-small). While some methods like Brier Score achieve slightly lower SCE values, SECA provides the best balance between accuracy and calibration without requiring hyper-parameter tuning.

| Methods | DBpedia | | | 20 Newsgroups | | |
|---|---|---|---|---|---|---|
| | BERT-small | | BERT-base | | BERT-small | | BERT-base |
| | TE \| SCE \| ECE \| AECE | TE \| SCE \| ECE \| AECE | TE \| SCE \| ECE \| AECE | TE \| SCE \| ECE \| AECE |
| Focal ($\gamma$=3.0) | 1.48 \| 10.9 \| 7.57 \| 7.56 | 1.43 \| 11.8 \| 8.10 \| 8.08 | 33.85 \| 11.4 \| 7.70 \| 7.88 | 38.06 \| 18.7 \| 5.34 \| 5.15 |
| LS ($\alpha$=0.1) | 1.28 \| 12.3 \| 8.37 \| 8.36 | 1.41 \| 13.3 \| 8.91 \| 8.90 | 31.51 \| 10.3 \| 5.81 \| 5.83 | 35.31 \| 13.4 \| 9.23 \| 10.2 |
| MMCE ($\beta$=4.0) | 1.71 \| 7.72 \| 5.33 \| 5.33 | 2.07 \| 8.61 \| 6.26 \| 6.26 | 32.71 \| 10.3 \| 4.66 \| 4.71 | 35.31 \| 16.0 \| 12.1 \| 12.1 |
| DCA ($\beta$=1.0) | 1.33 \| 0.94 \| 0.55 \| 0.56 | 1.42 \| 1.28 \| 0.72 \| 0.71 | 31.96 \| 15.3 \| 12.9 \| 12.9 | 35.41 \| 20.9 \| 18.1 \| 18.1 |
| FLSD ($\gamma$=3.0) | 1.48 \| 11.0 \| 7.61 \| 7.61 | 1.40 \| 12.0 \| 8.33 \| 8.32 | 33.61 \| 11.4 \| 7.65 \| 7.71 | 34.59 \| 13.5 \| 5.60 \| 5.49 |
| MDCA ($\gamma$, $\beta$=1.0) | 1.35 \| 1.77 \| 1.09 \| 1.07 | 1.35 \| 1.60 \| 0.80 \| 0.76 | 33.18 \| 13.2 \| 9.24 \| 9.31 | 33.24 \| 14.7 \| 11.1 \| 11.0 |
| Brier | 1.35 \| **0.85** \| 0.38 \| 0.43 | 1.47 \| **1.16** \| 0.66 \| 0.68 | 31.43 \| 11.4 \| 6.86 \| 7.03 | **32.39** \| 16.3 \| 13.4 \| 13.4 |
| OLS ($\alpha$=0.5) | 1.29 \| 6.74 \| 4.45 \| 4.39 | 1.42 \| 5.33 \| 3.42 \| 3.27 | **30.37** \| 12.4 \| 9.98 \| 9.89 | 32.60 \| 14.0 \| 8.43 \| 8.17 |
| DualFocal ($\gamma$=5.0) | 1.48 \| 9.78 \| 6.70 \| 6.70 | 1.48 \| 16.9 \| 11.7 \| 11.7 | 39.03 \| 10.8 \| 6.00 \| 5.88 | 42.02 \| 12.9 \| 7.44 \| 7.43 |
| AdaFocal | 1.37 \| 1.97 \| 1.24 \| 1.20 | 1.43 \| 2.42 \| 1.29 \| 1.26 | 39.29 \| 11.3 \| 5.97 \| 5.95 | 41.54 \| 19.1 \| 15.3 \| 15.2 |
| CE (baseline) | 1.32 \| 0.95 \| 0.53 \| 0.54 | 1.39 \| 1.17 \| 0.70 \| 0.71 | 32.39 \| 12.8 \| 9.36 \| 9.23 | 32.73 \| 20.0 \| 17.8 \| 17.8 |
| **SECA (Ours)** | **1.18** \| 0.97 \| **0.35** \| **0.38** | **1.38** \| 1.20 \| **0.63** \| **0.66** | 31.38 \| **8.89** \| **3.53** \| **3.89** | 32.73 \| **13.3** \| **5.33** \| **5.06** |

Table 3: Comparison between our method and other methods regarding calibration metrics: Test Error (%), SCE (‰), ECE (%) and AECE (%), for BERT-small/base on DBpedia and 20 Newsgroups datasets. A lower error is better. Results are averaged values based on five independent runs.

On the more challenging 20 Newsgroups dataset, SECA excels particularly in calibration metrics, achieving the lowest SCE (8.89‰ for BERT-small), ECE (3.53% and 5.33% for BERT-small and BERT-base, respectively), BERT-base, respectively), demonstrating substantial improvements over Cross-Entropy (e.g., reducing ECE from 9.36% to 3.53% on BERT-small, a 62% improvement).

## 4.3 Discussion

The experimental results across computer vision and natural language understanding tasks demonstrate that SECA effectively generalises across different modalities and architectures, providing robust calibration enhancement without requiring domain-specific tuning. Notably, SECA shows consistent improvements across datasets with varying numbers of classes, from dense class scenarios like CIFAR-10 (10 classes) and DBpedia (14 classes) to sparser class distributions like CIFAR-100 (100 classes) and ImageNet (1000 classes), indicating that its effectiveness stems from adaptive batch-level regularisation rather than dependence on specific class density conditions. This comprehensive evaluation validates SECA as an effective calibration method that consistently outperforms Cross-Entropy baselines across CNN, ViT, and BERT architectures. While some specialised methods may achieve better performance on specific metrics or datasets, SECA distinguishes itself through its hyper-parameter-free design, computational efficiency, and robust performance across diverse tasks. These characteristics, combined with its consistent calibration improvements and competitive accuracy, establish SECA as a practical alternative to conventional loss functions.

We provide comprehensive analyses in the appendix to further validate SECA's effectiveness and robustness. Specifically, we conducted ablation studies to investigate the impacts of varying batch sizes (Appendix D), demonstrate SECA's compatibility with post-hoc calibration techniques through integration with temperature scaling (Appendix E), present detailed reliability diagrams with 25-bin calibration analysis across all experimental configurations (Appendix F), and evaluate SECA's robustness via out-of-distribution detection experiments on CIFAR-10-C and SVHN (Appendix G).

## 5 CONCLUSION

We introduced SECA for self-guided model calibration, a novel train-time calibration method specifically designed to improve the reliability and trustworthiness of deep neural networks without extra hyper-parameters. SECA dynamically adjusts target distributions by leveraging batch-level averaged predictions, thereby encouraging well-calibrated probabilistic outputs through a self-guided regularisation mechanism. Extensive experiments demonstrated the effectiveness and versatility of SECA across diverse network architectures including CNNs, Vision Transformers, and BERT models, as well as various tasks spanning visual recognition and natural language understanding. SECA consistently improves model calibration on the majority of evaluated datasets and architectures, achieving substantially lower SCE, ECE, and Adaptive ECE compared to established baselines, without additional hyper-parameter tuning or significant computational overhead. Given its simplicity, hyper-parameter-free nature, computational efficiency, and broad applicability, SECA serves as a practical alternative to the Cross-Entropy loss for training well-calibrated neural networks.

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

## A EMPIRICAL STUDIES FOR THEORETICAL ANALYSIS

To complement the theoretical analysis discussed in Section 3.3, we provide a set of empirical studies that follow the same training settings as discussed in Section 4. These studies offer direct evidence for the core mechanisms underpinning SECA's calibration improvements, specifically from the perspectives of entropy regularisation, gradient dynamics, and knowledge distillation. We compare SECA with both the standard Cross-Entropy loss and Label Smoothing over the course of training, directly addressing the distinctions between SECA's adaptive calibration approach and Label Smoothing's fixed softening strategy. **Three key per-sample metrics** are tracked throughout training. **Average Entropy** is calculated from each sample's predicted probability distribution and measures the model's prediction calibration. Lower entropy indicates sharper predictions, while higher entropy suggests better-calibrated, less over-confident outputs. **Average KL Divergence** is computed between each sample's predicted probability and the average prediction of other samples in the batch that belong to the same class. It measures how well individual predictions align with the collective calibration signal of their class. **Average Cosine Similarity** measures how closely each sample's prediction aligns directionally with the corresponding batch-level average. It reflects the effectiveness of batch-level calibration regularisation in promoting well-calibrated outputs.

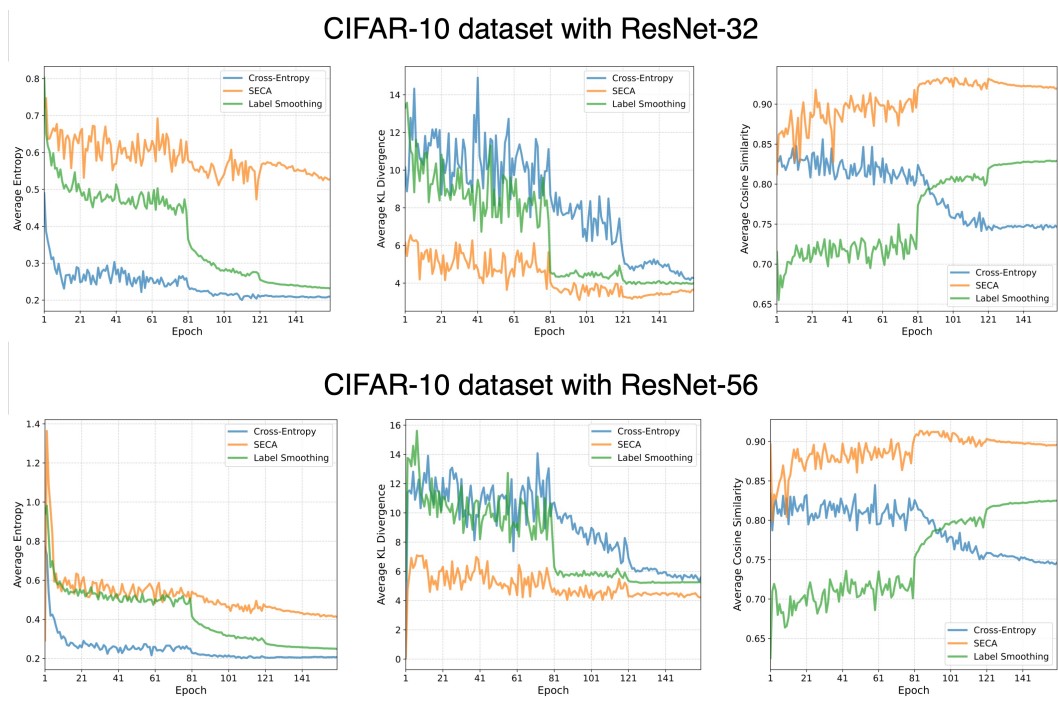

Figure 2: Comparison between SECA, Cross-Entropy, and Label Smoothing on CIFAR-10 dataset and ResNet models, with respect to three metrics: Average Entropy, Average KL Divergence, and Average Cosine Similarity.

As illustrated in Fig. 2 (top left and bottom left), on the CIFAR-10 dataset, both ResNet-32 and ResNet-56 models trained with SECA consistently exhibit higher average entropy than those trained with Cross-Entropy or Label Smoothing. This indicates that SECA promotes better-calibrated predictions by encouraging appropriate confidence levels, while avoiding the potential calibration inconsistencies observed with Label Smoothing in certain training phases. In addition, Fig. 2 (top centre and bottom centre) shows that the KL divergence between individual predictions and their corresponding batch-level class averages remains markedly lower under SECA compared to both baselines, particularly during the early and mid-training phases. This behaviour suggests that SECA effectively maintains prediction calibration by aligning individual outputs with class-informed expectations, thereby achieving better calibration than standard Cross-Entropy while providing more consistent regularisation than the fixed smoothing approach of Label Smoothing. Finally, as shown in Fig. 2 (top right and bottom right), the cosine similarity between per-sample predictions and their

corresponding batch-level averages is consistently higher under SECA than both Cross-Entropy and Label Smoothing, reinforcing the interpretation of SECA as operating through adaptive, class-aware calibration rather than uniform confidence adjustment.

### CIFAR-100 dataset with ResNet-32

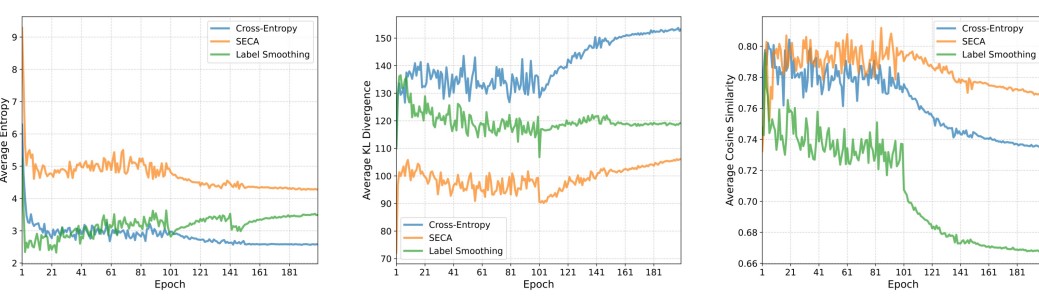

### CIFAR-100 dataset with ResNet-56

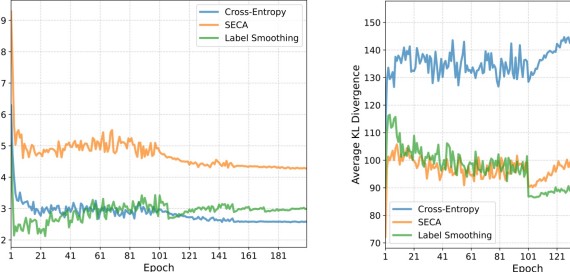 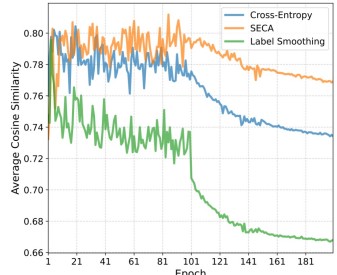

Figure 3: Comparison between SECA, Cross-Entropy, and Label Smoothing on the CIFAR-100 dataset and ResNet models, with respect to three metrics: Average Entropy, Average KL Divergence, and Average Cosine Similarity.

On the more complex CIFAR-100 dataset, the empirical advantages of SECA over both Cross-Entropy and Label Smoothing become even more pronounced. As shown in Fig. 3 (top left and bottom left), models trained with SECA demonstrate a noticeably larger average entropy compared to both baselines, which is particularly crucial given the challenge of achieving well-calibrated predictions across a larger number of classes. This enhanced entropy demonstrates SECA's effectiveness in promoting appropriate calibration across multi-class scenarios where Label Smoothing's uniform approach may prove insufficient. Notably, Fig. 3 (top centre and bottom centre) shows that the KL divergence under SECA remains substantially lower across all epochs compared to both Cross-Entropy and Label Smoothing, suggesting that SECA maintains superior class-consistent calibration and reduces the prediction instability observed with alternative approaches. Additionally, as illustrated in Fig. 3 (top right and bottom right), cosine similarity remains consistently higher under SECA than both baselines, confirming that SECA's adaptive mechanism enables predictions to converge more effectively toward well-calibrated batch-level expectations, outperforming the fixed regularisation strategies of Label Smoothing.

Similar calibration advantages are observed in the natural language understanding tasks (Fig. 4 and Fig. 5). The empirical analysis across four datasets and multiple architectures confirms that SECA consistently exhibits superior calibration behaviour compared to both Cross-Entropy and Label Smoothing, validating its theoretical foundations (as discussed in Section 3.3). The observed increase in average entropy supports SECA's role as an adaptive calibration regulariser that dynamically adjusts prediction confidence based on batch-level class information. The substantial reduction in KL divergence demonstrates SECA's effectiveness in promoting well-calibrated gradients and maintaining appropriate prediction confidence across diverse tasks. Additionally, the higher cosine similarity between predictions and batch-level averages confirms SECA's adaptive calibration mechanism, whereby individual predictions are dynamically aligned with the calibrated consensus of their class peers, surpassing the static approach of Label Smoothing.

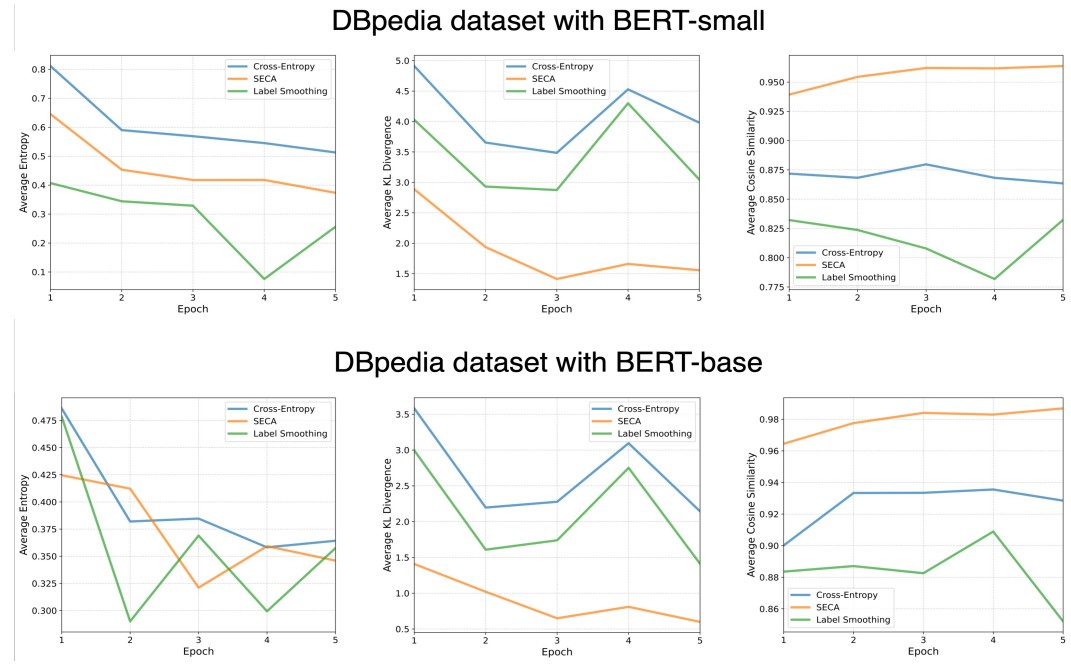

Figure 4: Comparison between SECA, Cross-Entropy, and Label Smoothing on DBpedia dataset and BERT models, with respect to three metrics: Average Entropy, Average KL Divergence, and Average Cosine Similarity.

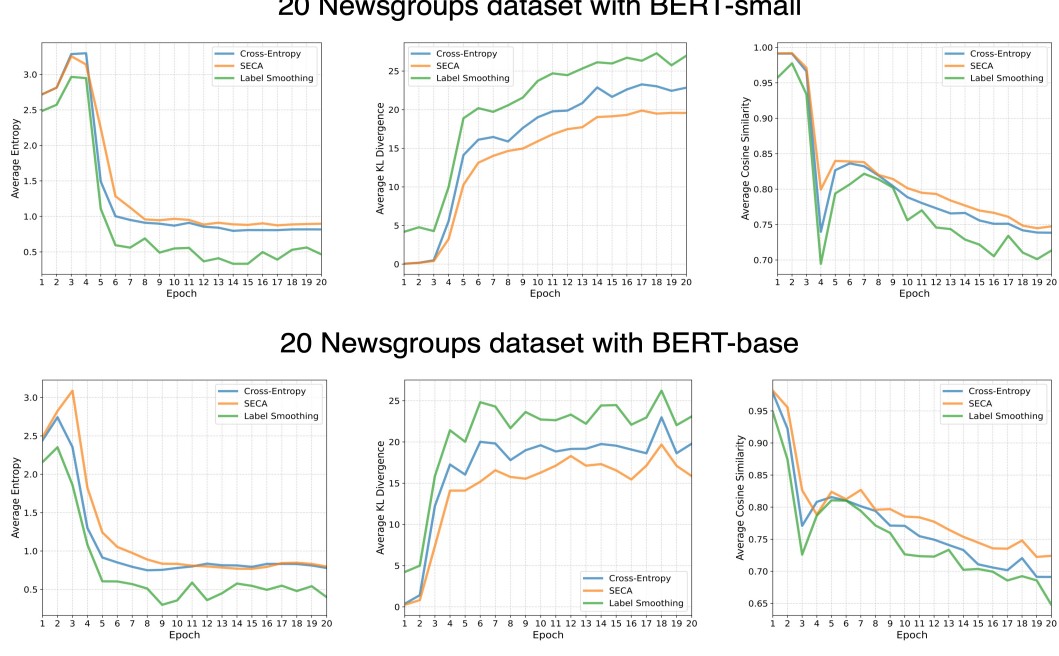

Figure 5: Comparison between SECA, Cross-Entropy, and Label Smoothing on 20 Newsgroups dataset and BERT models, with respect to three metrics: Average Entropy, Average KL Divergence, and Average Cosine Similarity.

## B  GENERAL TRAINING HYPER-PARAMETERS

Table 4 presents the experimental configuration for each dataset–model pairing employed in our SECA evaluation. These hyper-parameters are held constant across all comparison methods to ensure fair evaluation of calibration performance.

| Dataset | Model | #Classes | LR | LR Scheduler | Batch Size | Epochs |
|---------|-------|----------|-----|--------------|------------|--------|
| CIFAR-10 | ResNet32 | 10 | 0.1 | MultiStep | 128 | 160 |
| CIFAR-10 | ResNet56 | 10 | 0.1 | MultiStep | 128 | 160 |
| CIFAR-100 | ResNet32 | 100 | 0.1 | MultiStep | 128 | 200 |
| CIFAR-100 | ResNet56 | 100 | 0.1 | MultiStep | 128 | 200 |
| ImageNet | ViT-small | 1000 | 5e-4 | Cosine+Warmup (0.02) | 1024 | 300 |
| DBpedia | BERT-small | 14 | 3e-5 | Linear+Warmup (0.1) | 256 | 5 |
| DBpedia | BERT-base | 14 | 2e-5 | Linear+Warmup (0.1) | 128 | 5 |
| 20 Newsgroups | BERT-small | 20 | 3e-5 | Linear+Warmup (0.1) | 256 | 20 |
| 20 Newsgroups | BERT-base | 20 | 2e-5 | Linear+Warmup (0.1) | 128 | 20 |

Table 4: Dataset–model pairs and their corresponding hyper-parameter setups, including learning rate, scheduler type, batch size, and number of training epochs.

## C  EVALUATION METRICS

To quantify calibration, we use several standard metrics in this study:

**Expected Calibration Error (ECE)** Guo et al. (2017) measures the average discrepancy between confidence and accuracy across bins of predictions grouped by confidence, defined as follows:

$$\text{ECE} = \sum_{m=1}^{M} \frac{|B_m|}{n} \left| \text{acc}(B_m) - \text{conf}(B_m) \right|, \tag{16}$$

where $n$ is the total number of samples, $M$ is the number of confidence bins, $B_m$ is the set of indices of samples whose predicted confidence falls into $m$-th bin. Accordingly, $\text{acc}(B_m)$ is the average accuracy in bin $m$, $\text{conf}(B_m)$ is the average predicted confidence in bin $m$.

**Static Calibration Error (SCE)** Nixon et al. (2019) computes the calibration error per-class independently and averaged error across all classes. This making the calibration for each class is considered equally, avoiding domination by majority classes, defined as follows:

$$\text{SCE} = \frac{1}{C} \sum_{c=1}^{C} \sum_{m=1}^{M} \frac{|B_{m,c}|}{n_c} \left| \text{acc}(B_{m,c}) - \text{conf}(B_{m,c}) \right|, \tag{17}$$

where $C$ denotes the number of classes, $n_c$ is the number of samples belonging to class $c$, $B_{m,c}$ is the set of samples of class $c$ whose confidence falls into bin $m$. $\text{acc}(B_{m,c})$ and $\text{conf}(B_{m,c})$ are the average accuracy and confidence for class $c$ in bin $m$, respectively.

**Adaptive ECE (AECE)** Ding et al. (2020) improves upon ECE by adjusting bin sizes based on the density of confidence scores, ensuring each bin contains approximately the same number of samples. This reduces bias from uneven sample distribution across confidence ranges.

$$\text{AECE} = \sum_{m=1}^{M} \frac{|B_m|}{n} \left| \text{acc}(B_m) - \text{conf}(B_m) \right|, \tag{18}$$

Specifically, SCE evaluates calibration error independently for each class and averages these values, ensuring class-level calibration fairness. ECE measures the overall alignment between predicted confidence and empirical accuracy, providing a global assessment of calibration quality. AECE adaptively adjusts bin sizes based on the confidence distribution, making it robust to unevenly distributed confidence scores.

# D ABLATION STUDIES ON THE IMPACTS OF BATCH SIZE

As shown in Fig. 6, SECA demonstrates substantial improvements in calibration metrics (SCE, ECE, and AECE) compared to the Cross-Entropy baseline across all tested batch sizes. Notably, the calibration benefits of SECA become increasingly pronounced with larger batch sizes, underscoring the efficacy of leveraging richer batch-level distributional statistics for adaptive target construction.

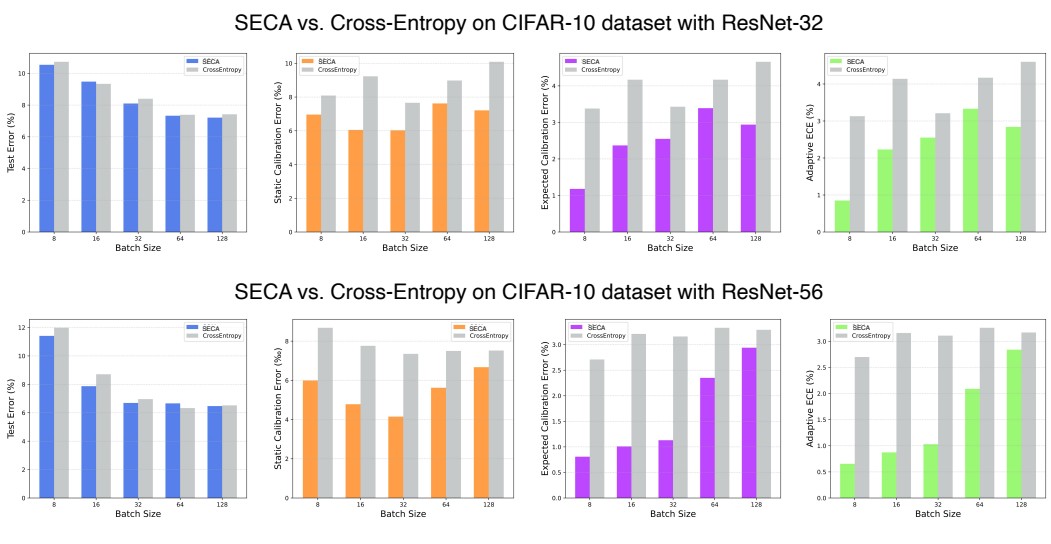

Figure 6: Comparison between SECA and baseline Cross-Entropy loss on the CIFAR-100 dataset and ResNet models, with respect to varying batch sizes from 8 to 128.

As shown in Fig. 7, SECA consistently outperformed Cross-Entropy across all tested batch sizes, as well as achieving lower SCE, ECE, and AECE values. Notably, as batch size increases, both SECA and Cross-Entropy yield lower test error rates, but SECA typically maintains its calibration advantage, indicating its stable performance over the standard Cross-Entropy loss.

Figure 7: Comparison between SECA and baseline Cross-Entropy loss on CIFAR-10 dataset and ResNet models, with respect to varying batch sizes from 8 to 128.

For the DBpedia dataset (shown in Fig. 8), SECA shown consistent superiority over the Cross-Entropy for both BERT-small and BERT-base architectures across all batch sizes. Particularly for

larger batch sizes, SECA not only maintained lower calibration errors but also exhibited improved robustness, suggesting its efficacy on Transformer networks and natural language tasks.

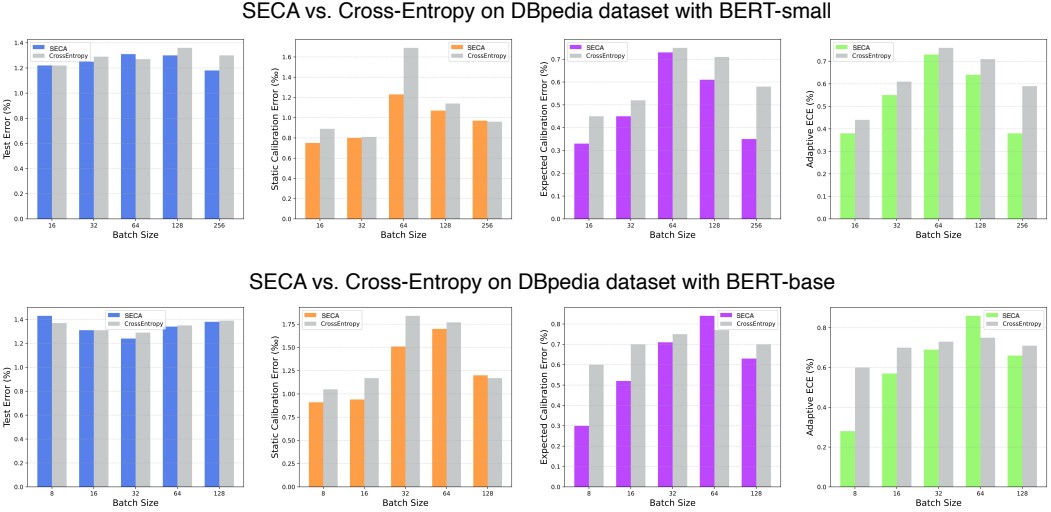

Figure 8: Comparison between SECA and baseline Cross-Entropy loss on DBpedia dataset and BERT models, with respect to varying batch sizes from 16 to 256.

As illustrated in Fig. 9, the results on the 20 Newsgroups dataset further support above observations. SECA consistently achieved superior calibration performance across different batch sizes. The calibration gap between SECA and Cross-Entropy widened with increasing batch sizes, demonstrating the robustness and effectiveness of its adaptive batch-level softening in textual scenarios.

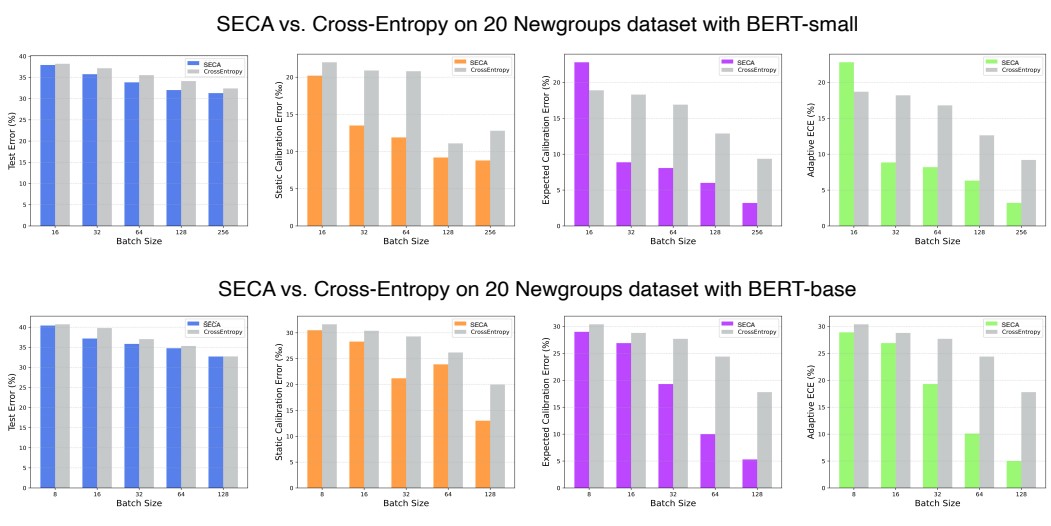

Figure 9: Comparison between SECA and baseline Cross-Entropy loss on 20 Newsgroups dataset and BERT models, with respect to varying batch sizes from 16 to 256.

Overall, these ablation studies show that although SECA naturally inherits a degree of batch-size dependence due to its use of batch-level class-wise statistics, it remains consistently superior to Cross-Entropy across all tested batch sizes and across all datasets and architectures. The modest fluctuations observed in certain configurations, such as larger batch sizes on CIFAR-10, do not change its overall advantage. Importantly, SECA does not require tuning of the batch size to obtain

strong results, all main experiments employ standard batch sizes commonly used in prior work, under which SECA remains stable and effective. We acknowledge that extreme batch sizes may affect calibration strength, and we view this as an inherent limitation shared by batch-dependent calibration methods. In practice, SECA provides reliable behaviour under typical training setups.

## E  INTEGRATION WITH POST-HOC CALIBRATION

| Methods | CIFAR-10 | | | | CIFAR-100 | | | |
|---|---|---|---|---|---|---|---|---|
| | ResNet32 | | ResNet56 | | ResNet32 | | ResNet56 | |
| | Pre T | Post T | Pre T | Post T | Pre T | Post T | Pre T | Post T |
| Focal ($\gamma$=3.0) | 4.54 | 2.91 (0.8) | 4.46 | 2.01 (0.8) | 2.26 | 1.92 (1.1) | 2.07 | 2.07 (1.0) |
| LS ($\alpha$=0.1) | 6.37 | 2.07 (0.8) | 5.49 | 1.70 (0.8) | 2.80 | 2.80 (0.9) | 2.35 | 2.35 (1.0) |
| MMCE ($\beta$=4.0) | 3.40 | 3.04 (0.6) | 3.30 | 2.24 (0.6) | 7.53 | 1.97 (1.2) | 6.93 | 1.34 (1.2) |
| DCA ($\beta$=1.0) | 4.24 | 2.95 (1.8) | 3.34 | 2.39 (1.8) | 12.0 | 1.20 (1.6) | 11.1 | 2.22 (1.7) |
| FLSD ($\gamma$=3.0) | 4.44 | 1.70 (0.9) | 4.75 | 1.70 (0.8) | 2.16 | 3.62 (1.1) | 2.30 | 2.77 (1.1) |
| MDCA ($\gamma, \beta$=1.0) | 1.84 | 0.82 (1.2) | **1.25** | **0.45 (1.2)** | 5.61 | 1.94 (1.2) | 5.24 | 1.86 (1.3) |
| Brier | 2.61 | 1.27 (1.2) | 2.15 | 1.03 (1.2) | 5.56 | 2.07 (1.2) | 4.94 | 1.98 (1.2) |
| OLS ($\alpha$=0.5) | 3.31 | 2.73 (1.4) | 2.80 | 1.69 (1.3) | 4.51 | 1.64 (1.2) | 2.44 | 1.31 (1.2) |
| DualFocal ($\gamma$=5.0) | **1.82** | **0.60 (0.9)** | 2.61 | 0.54 (0.8) | 3.30 | 1.81 (1.1) | 1.94 | 1.56 (1.1) |
| AdaFocal | 2.69 | 0.90 (1.2) | 1.44 | 0.51 (1.1) | 3.42 | 2.09 (1.3) | 2.75 | 1.83 (1.3) |
| CE (baseline) | 3.86 | 2.28 (1.9) | 3.10 | 1.75 (1.6) | 10.0 | 2.85 (1.6) | 9.09 | 2.52 (1.3) |
| **SECA (Ours)** | 2.94 | 2.53 (1.3) | 2.41 | 1.50 (1.2) | **1.90** | **1.10 (1.1)** | **1.71** | **1.25(1.2)** |

Table 5: Expected Calibration Error (%) before and after Temperature Scaling on CIFAR-10 and CIFAR-100 datasets. Numbers in parentheses indicate the optimal temperature values learned during calibration. Lower ECE values indicate better calibration performance. Temperature values $\approx 1.0$ suggest innately well-calibrated models.

As shown in the Table 5, the integration of temperature scaling with various calibration methods provides critical insights into the intrinsic calibration properties of different approaches. Our proposed SECA method demonstrates superior performance in the post-hoc calibration setting, achieving the lowest ECE values on CIFAR-100 for both ResNet32 (1.10%) and ResNet56 (1.25%) architectures. Notably, SECA consistently requires minimal temperature adjustment (T $\approx$ 1.1–1.3), indicating that it produces inherently well-calibrated predictions that require minimal post-hoc correction.

The effectiveness of temperature scaling varies significantly across methods and datasets. While most approaches benefit from temperature scaling, the magnitude of improvement differs substantially. The Cross-Entropy baseline exhibits the largest temperature adjustments (T $\approx$ 1.3–1.9), reflecting significant initial over-confidence that is effectively mitigated through scaling, resulting in ECE reductions of up to 74% on CIFAR-100. In contrast, methods such as DualFocal and SECA require more modest temperature adjustments, suggesting better initial calibration properties.

Several methods demonstrate exceptional synergy with temperature scaling. DualFocal achieves the best post-TS performance on CIFAR-10 ResNet32 (0.60% ECE), while MDCA attains remarkable calibration on CIFAR-10 ResNet56 (0.45% ECE). However, some methods show diminishing returns or inconsistent improvements. Label Smoothing exhibits minimal post-TS improvement on CIFAR-100, with ECE remaining unchanged at 2.35% for ResNet56, potentially indicating over-regularisation that limits the effectiveness of subsequent temperature scaling.

CIFAR-100 results consistently demonstrate larger absolute improvements from temperature scaling compared to CIFAR-10, reflecting the increased calibration challenges associated with fine-grained classification tasks. Despite this increased complexity, SECA maintains robust performance across both datasets, demonstrating its effectiveness in diverse classification scenarios. The learned temperature values serve as diagnostic indicators of initial model calibration, with values substantially greater than 1.0 indicating over-confidence requiring correction, while values approaching 1.0 suggest well-calibrated base models.

# F RELIABILITY ANALYSIS

The reliability diagrams presented in Figures 10, 11, 12, and 13 provide a comprehensive visual assessment of model calibration across different calibration methods using 25 confidence bins. Each subplot depicts the alignment between expected confidence (red bars) and actual accuracy (purple bars), with the diagonal dashed line representing perfect calibration. The analysis reveals several key insights regarding the calibration behaviour of various methods.

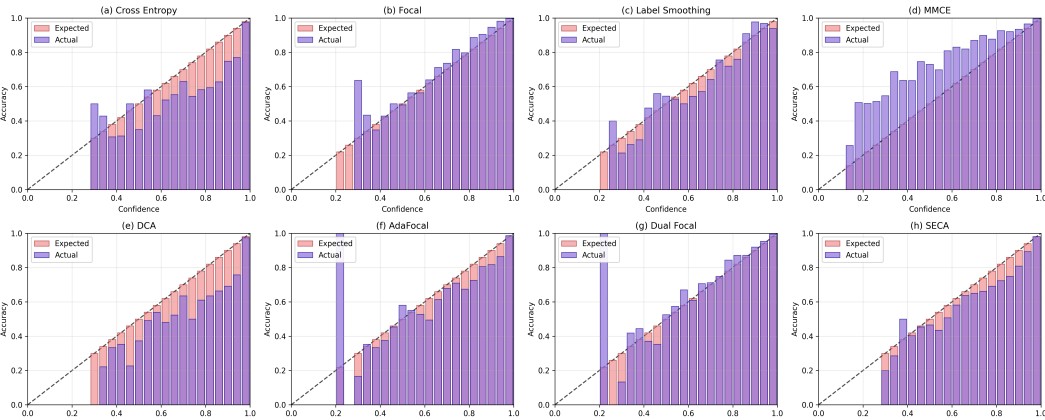

Figure 10: Reliability diagrams on CIFAR-10 with ResNet-32.

**CIFAR-10 Results.** On CIFAR-10 with ResNet-32 (Figure 10), SECA demonstrates superior calibration performance across the confidence spectrum. While baseline methods such as Cross Entropy exhibit substantial over-confidence, particularly in the high-confidence bins (0.8–1.0) where the majority of predictions concentrate, SECA maintains close alignment between predicted confidence and actual accuracy.

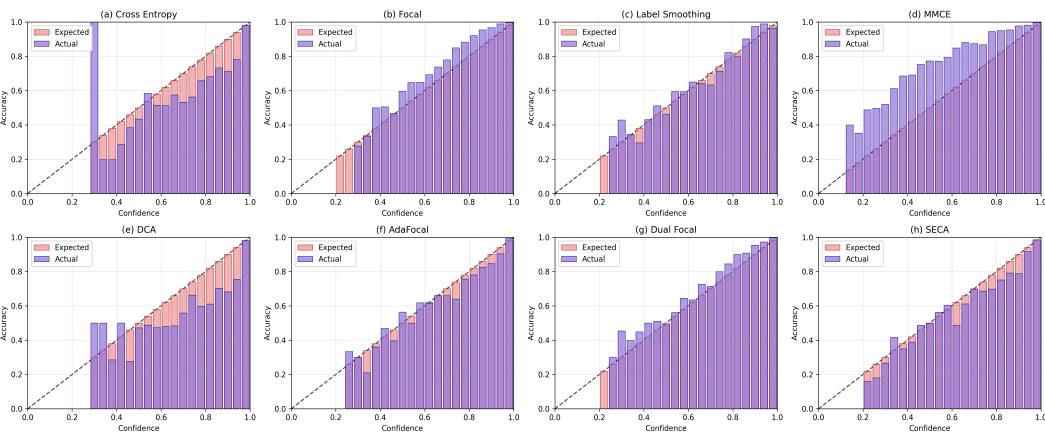

Figure 11: Reliability diagrams on CIFAR-10 with ResNet-56.

The trend continues with ResNet-56 (Figure 11). The reliability diagrams reveal that traditional calibration methods such as Focal Loss and MMCE struggle to maintain consistent calibration across all confidence bins, often exhibiting erratic behaviour in mid-range confidence intervals (0.4–0.8). In contrast, SECA maintains smooth and consistent calibration behaviour, with minimal deviation from the perfect calibration line across all 25 bins.

**CIFAR-100 Results.** The superiority of SECA becomes even more pronounced on the challenging CIFAR-100 dataset, which presents increased complexity due to its larger number of classes and

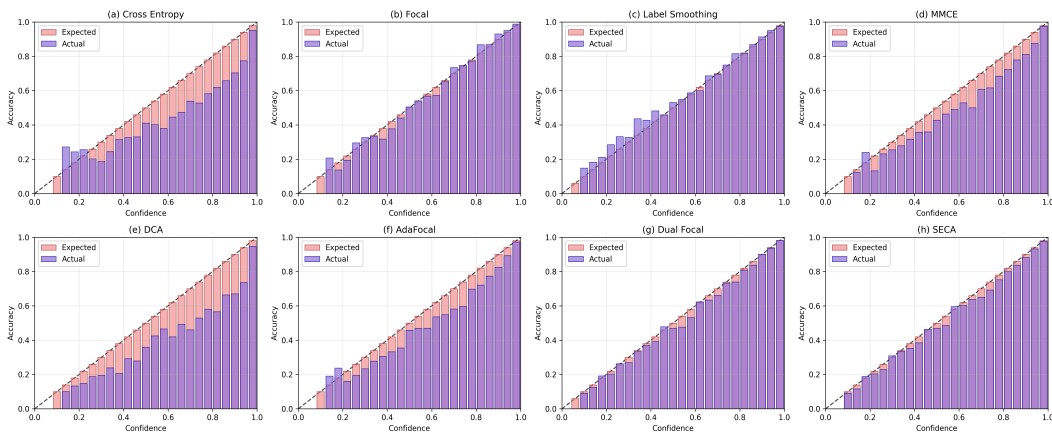

Figure 12: Reliability diagrams on CIFAR-100 with ResNet-32.

reduced samples per class. On ResNet-32 (Figure 12), SECA achieves remarkable calibration with an ECE of 1.90%, while Cross Entropy exhibits severe over-confidence with an ECE of 10.0%—a five-fold improvement. The reliability diagrams clearly illustrate this disparity, Cross Entropy shows substantial gaps between expected confidence and actual accuracy across multiple bins, while SECA maintains near-perfect alignment.

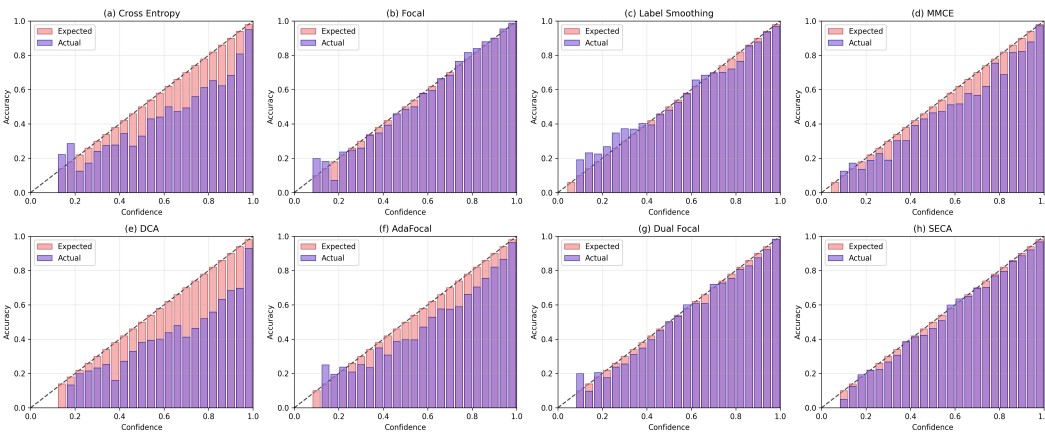

Figure 13: Reliability diagrams on CIFAR-100 with ResNet-56.

Similarly, with ResNet-56 (Figure 13), SECA sustains excellent calibration performance (ECE: 1.71%) compared to Cross Entropy (ECE: 9.09%). The fine-grained 25-bin analysis reveals that SECA's calibration improvements are not merely concentrated in specific confidence ranges but are consistently maintained across the entire confidence spectrum. This uniform improvement is particularly evident in the medium-confidence bins (0.3–0.8), where many competing methods exhibit significant calibration errors.

## G    ROBUSTNESS ON DATASET SHIFT

We evaluate the robustness of our proposed SECA method against dataset shift by conducting out-of-distribution (OoD) detection experiments. Following established protocols (Mukhoti et al., 2020; Tao et al., 2023), we train models on CIFAR-10 as the in-distribution dataset and evaluate OoD detection performance using CIFAR-10-C (corrupted images) (Hendrycks & Dietterich, 2018) and SVHN (Street View House Numbers) (Goodfellow et al., 2013) as out-of-distribution datasets. The AUROC metric is employed to measure detection performance, with higher values indicating better separation between in-distribution and out-of-distribution samples.

We conduct experiments on two ResNet architectures (ResNet32 and ResNet56) and compare SECA against ten state-of-the-art calibration methods (similar to Section 4.1). For each method, we report AUROC scores both before (Pre-T) and after (Post-T) temperature scaling to assess the impact of post-hoc calibration on OoD detection capabilities.

| Methods | CIFAR-10-C | | | | SVHN | | | |
| | ResNet32 | | ResNet56 | | ResNet32 | | ResNet56 | |
| | Pre T | Post T | Pre T | Post T | Pre T | Post T | Pre T | Post T |
|---|---|---|---|---|---|---|---|---|
| Focal ($\gamma$=3.0) | 81.36 | 80.48 | 88.54 | 87.61 | 86.60 | 86.09 | 89.18 | 88.01 |
| LS ($\alpha$=0.1) | 66.19 | 66.86 | 88.54 | 87.61 | 60.83 | 62.99 | 89.18 | 87.61 |
| MMCE ($\beta$=4.0) | 86.34 | 85.59 | 85.81 | 85.13 | 97.13 | 85.59 | 95.50 | 95.01 |
| DCA ($\beta$=1.0) | 83.91 | 84.84 | 83.06 | 83.66 | 90.83 | 91.49 | 90.47 | 91.30 |
| FLSD ($\gamma$=3.0) | 80.08 | 79.82 | 88.16 | 87.62 | 85.31 | 85.10 | 87.37 | 87.18 |
| MDCA ($\gamma, \beta$=1.0) | 83.55 | 83.62 | 86.49 | 86.78 | 87.83 | 87.97 | 83.84 | 84.34 |
| Brier | 78.19 | 78.36 | 83.54 | 83.74 | 90.46 | 90.70 | 89.91 | 90.20 |
| OLS ($\alpha$=0.5) | 80.21 | 80.42 | 83.47 | 83.58 | 87.45 | 87.90 | 90.24 | 90.51 |
| DualFocal ($\gamma$=5.0) | 78.76 | 78.63 | 88.34 | 87.84 | 90.41 | 90.27 | 89.45 | 89.05 |
| AdaFocal | 87.47 | 87.63 | 89.72 | 89.90 | 86.70 | 86.67 | 88.39 | 88.51 |
| CE (baseline) | 81.15 | 81.23 | 88.64 | 89.00 | 86.14 | 85.56 | 92.22 | 92.80 |
| **SECA (Ours)** | 83.28 | 83.34 | 88.69 | 88.64 | 89.46 | 89.29 | 93.06 | 93.13 |

Table 6: Robustness evaluation on dataset shift. AUROC (%) for ResNet-32/56 models trained on CIFAR-10 (in-distribution) and evaluated on CIFAR-10-C (Gaussian noise, severity 5) and SVHN (out-of-distribution). Pre T/Post T indicate results before/after temperature scaling.

As shown in Table 6, SECA demonstrates exceptional OoD detection capabilities across both evaluation scenarios. On the SVHN dataset, SECA achieves the highest AUROC scores among all methods, reaching 93.06% with ResNet56 and 89.46% with ResNet32. For CIFAR-10-C detection, SECA consistently ranks among the top-performing methods, achieving 88.69% (ResNet56) and 83.28% (ResNet32), representing substantial improvements of +2.13% and +3.32% over the Cross-Entropy baseline, respectively.

A critical advantage of SECA lies in its remarkable stability when temperature scaling is applied. While some calibration methods exhibit significant degradation, SECA maintains virtually unchanged performance. The method exhibits minimal variations of ±0.17% or less across all experimental conditions, demonstrating that calibration improvements come without sacrificing OoD detection capabilities.

In terms of cross-architecture performance, SECA demonstrates robust performance across different network architectures. Unlike some methods that show inconsistent behaviour between ResNet32 and ResNet56, SECA maintains strong and consistent performance improvements across both architectures, indicating good generalisation properties.

# H    IMBALANCE DATASET

| Methods | Imbalance CIFAR-100 | | |
|---|---|---|---|
| | IF-10 | IF-50 | IF-100 |
| Focal ($\gamma$=3.0) | 3.83 | 5.78 | 6.56 |
| LS ($\alpha$=0.1) | 3.53 | 5.89 | 6.92 |
| MMCE ($\beta$=4.0) | 5.40 | **5.36** | **6.19** |
| DCA ($\beta$=1.0) | 5.10 | 7.63 | 8.48 |
| FLSD ($\gamma$=3.0) | 3.74 | 5.89 | 6.80 |
| MDCA ($\gamma, \beta$=1.0) | 4.27 | 6.51 | 7.00 |
| Brier | 6.06 | 9.22 | 9.67 |
| OLS ($\alpha$=0.5) | 3.55 | 5.72 | 6.61 |
| DualFocal ($\gamma$=5.0) | 3.57 | 5.56 | 6.43 |
| AdaFocal | 4.15 | 6.43 | 7.38 |
| CE (baseline) | 4.47 | 7.15 | 8.25 |
| SECA (Ours) | **3.45** | 5.46 | 6.42 |

Table 7: Calibration performance (SCE (‰)) on imbalanced CIFAR-100 under varying imbalance factors (IF-10/50/100) as suggested by Cui et al. (2019). SECA consistently achieves the best or second-best SCE across all imbalance levels, demonstrating its robustness in long-tailed settings without requiring imbalance-specific tuning.

Table 7 reports SCE performance on imbalanced CIFAR-100 with imbalance factors of 10, 50, and 100. SECA exhibits consistently strong calibration behaviour across all imbalance levels. It achieves the lowest SCE under IF-10, and remains highly competitive under IF-50 and IF-100. Importantly, **SECA requires no adjustment or imbalance-specific hyper-parameters**, yet maintains calibration quality even when minority classes become severely underrepresented. In contrast, several existing methods (e.g., DCA, AdaFocal) deteriorate notably as imbalance increases. These results support our claim that SECA's batch-conditional hybrid targets naturally extend to long-tailed data and remain effective even when per-class sample frequency is highly uneven.

