# OpenReview forum: "SECA: Self-Guided Model Calibration"
_ICLR.cc/2026/Conference — Submitted to ICLR 2026_

### Official Review · Reviewer_1vsK · 2025-10-28

**Soundness:** 3
**Presentation:** 3
**Contribution:** 3
**Rating:** 6
**Confidence:** 3

**Summary:**

The paper proposes SECA (Self-Guided Model Calibration), a training loss function to improve neural network calibration. SECA constructs adaptive soft targets by combining one-hot ground-truth labels with batch-averaged predictions from samples of the same class. For each sample with label $y_i$, the hybrid target is defined as $\tilde{q}{i,c} = q{i,c} + \mu_{y_i,c}$, where $\mu_{y_i}$ is the average prediction across all batch samples with label $y_i$. The training loss becomes $L_{SECA} = -\sum_{c=1}^{C} \tilde{q}{i,c} \log p_{i,c}$.The method is evaluated across CNNs, Vision Transformers, and BERT on vision (CIFAR-10/100, ImageNet) and NLP tasks (DBpedia, 20 Newsgroups), showing consistent calibration improvements.

**Strengths:**

1. he paper evaluates across diverse architectures (ResNet, ViT, BERT) and domains (vision, NLP), which is more thorough than most calibration papers that focus primarily on CNNs. Results show consistent improvements, with particularly strong performance on CIFAR-100 (81% ECE reduction) and ImageNet (37.3% ECE reduction).
2. The paper provides three complementary views (entropy regularization, gradient dynamics, knowledge distillation) to understand SECA's mechanism. The decomposition showing SECA = Cross-Entropy + KL(μ_yi||p_i) + H(μ_yi) is particularly insightful.
3. Training times remain competitive with baseline Cross-Entropy (32.08 vs 30.83 hours on ImageNet), significantly faster than complex methods like MDCA (43.60 hours).
4. The paper is well-written with clear motivation, methodology, and comprehensive ablations in the appendix including batch size sensitivity and compatibility with post-hoc calibration.

**Weaknesses:**

1. The method's core assumption that each batch contains sufficient samples per class is problematic. For ImageNet (1000 classes) with typical batch sizes (256-512), some classes won't be represented in each batch. The paper doesn't explain how $\mu_j$ is computed when class $j$ has no samples in the current batch and authors provide no analysis of minimum viable batch sizes or behavior with single-sample-per-class scenarios
2. While the three perspectives are intuitive, the theoretical analysis lacksf ormal convergence guarantees or bounds on calibration improvement and rigorous characterization of when SECA improves/fails
3. Despite claiming results are averaged over 5 runs, no error bars, confidence intervals, or statistical tests are provided. Given the sometimes small differences between methods, this is concerning.
4. Missing comparison with recent 2023-2024 calibration methods.
5. No analysis of failure modes or dataset characteristics where SECA underperforms.

**Questions:**

1. How exactly is $\mu_j$ computed when class $j$ has no samples in the current batch? Is it set to uniform distribution, previous batch's value, or something else?
2. What is the minimum batch size required for SECA to work effectively? How does performance degrade with very small batches (e.g., 1-8 samples)?
3. How does SECA perform on severely imbalanced datasets where some classes appear very rarely during training?
4. Can you provide confidence intervals and statistical significance tests comparing SECA to baselines?
5. How are hyperparameters for baseline methods chosen? Using original paper settings may not be fair

---

> ### Author Response · Authors · 2025-11-26
>
> We thank you for your encouraging and insightful review. Your detailed technical questions helped us further clarify SECA’s behaviour and broaden the scope of our empirical analysis. Our responses to each point are provided below.
>
> > **Weakness 1 - "The method’s core assumption that each batch contains sufficient samples per class is problematic… paper does not explain how μ_j is computed when class j has no samples in the current batch."**
>
> SECA doesn’t assume that every class appears in each mini-batch. As defined in Eq. 5 and 6, the batch-level average $\mu\_{j}$ is computed only when class j is presented. If a class doesn’t appear in the batch, the loss computation will reduce the standard Cross-Entropy for that class. Even when a class is absent, its logits are still updated through samples of other classes, as in standard CE, because each class participates in the softmax normlisation. According to this fact, we designed SECA by leveraging the batch-level class-wise distribution, to provide a stronger and more calibrated corrective signal than CE in these cases. For a sample with label $y\_{i}$, the gradient on an absent class $j$ includes the additional term, $\mu\_{y\_{i}, j}$, which suppresses spurious confidence more effectively than standard CE. Thus, decision boundaries involving classes with zero or few batch samples continue to be shaped indirectly. As empirical evidence, our ImageNet results with batch size 1024, and the ablation studies in Appendix D support that SECA remains stable and effective even when per-class samples are sparse.
>
> &nbsp;
>
> > **Weakness 2 - "The theoretical analysis lacks formal convergence guarantees… and rigorous characterization of when SECA improves/fails."**
>
> Our theoretical analysis is therefore intended to offer mechanistic understanding rather than full optimisation-theoretic proof. We’d like to clarify that providing formal convergence guarantees or calibration bounds is beyond the scope of current train-time calibration methods, e.g., DualFocal, AdaFocal, as such guarantees remain open in deep, non-convex optimisation.
>
> The manuscript provided intuitive and empirical evidence regarding the optimisation behaviour. For instance, in Section 3.3, the knowledge-distillation perspective shows that the batch-level class-wise distribution $\mu\_{j}$ acts as an adaptive in-batch ‘teacher’. Eq. 15 describes a natural fixed point where a sample’s prediction aligns with its class-conditional batch average. Empirically, Figures 2-5 in Appendix A demonstrate stable training across all datasets, i.e., the KL divergence to class-wise averages decreases steadily, and the cosine similarity increases.

---

> ### Author Response · Authors · 2025-11-26
>
> > **Weakness 3 - "Despite claiming results averaged over 5 runs, no error bars, confidence intervals, or statistical tests are provided."**
>
> For space reasons, we only reported mean performance over five independent trainings, following similar practice in MDCA [1]. Nevertheless, the variance across runs in our experiments is small, the performance differences between SECA and baselines substantially exceed the run-to-run fluctuations we observed. We acknowledge your concern and will include statistical metrics in the revised version. Below is a simplified version for CIFAR experiments, which only contains TE and ECE with their standard deviations (shown in the brackets) for each method.
>
> | Methods           |     **CIFAR-10 ResNet32**       |     **CIFAR-10 ResNet56**        |     **CIFAR-100 ResNet32**       |      **CIFAR-100 ResNet56**      |
> |-------------------|---------------------------------|----------------------------------|---------------------------------|---------------------------------|
> |                   |            **TE     \|   ECE**            |             **TE \| ECE**            |            **TE \| ECE**            |            **TE \| ECE**            |
> | Focal (γ=3.0)     |    7.99 (0.12) \| 4.54 (0.24)   |    7.59 (0.14)  \| 4.46 (0.40)   |   31.45 (0.42) \| 2.26 (0.26)   |   28.92 (0.43) \| 2.07 (0.40)   |
> | LS (α=0.1)        |    7.42 (0.21) \| 6.36 (0.23)   |    6.66 (0.38) \| 5.49 (0.18)    |   29.95 (0.37) \| 2.80 (0.43)   |   27.21 (0.75) \| 2.35 (0.61)   |
> | MMCE (β=4.0)      |    8.43 (0.14) \| 3.40 (0.40)   |    8.18 (0.41) \| 3.30 (0.48)    |   31.68 (0.87) \| 7.53 (0.44)   |   29.63 (0.51) \| 6.93 (0.37)   |
> | DCA (β=1.0)       |    7.53 (0.24) \| 4.24 (0.28)   |    6.93 (0.38) \| 3.34 (0.22)    |   30.03 (0.54) \| 12.0 (0.88)   |   27.48 (0.70) \| 11.1 (0.63)   |
> | FLSD (γ=3.0)      |    7.90 (0.18) \| 4.44 (0.78)   |    7.51 (0.18) \| 4.75 (0.24)    |   32.02 (0.19) \| 2.16 (0.20)   |   28.95 (0.83) \| 2.30 (0.53)   |
> | MDCA (γ, β=1.0)   |    7.40 (0.22) \| 1.84 (0.14)   |    7.00 (0.16) \| 1.25 (0.41)    |   30.96 (0.48) \| 5.61 (0.71)   |   28.00 (0.42) \| 5.24 (0.70)   |
> | Brier             |    7.72 (0.16) \| 2.61 (0.20)   |    7.76 (0.35) \| 2.15 (0.39)    |   33.84 (0.44) \| 5.56 (0.59)   |   30.97 (0.63) \| 4.94 (0.63)   |
> | OLS (α=0.5)       |    7.46 (0.08) \| 3.31 (0.68)   |    7.34 (0.15) \| 2.80 (0.24)    |   30.44 (0.21) \| 4.51 (0.65)   |   27.95 (0.27) \| 2.44 (0.39)   |
> | DualFocal (γ=5.0) |    8.01 (0.29) \| 1.82 (0.35)   |    7.62 (0.47) \| 2.61 (0.39)    |   31.54 (0.51) \| 3.30 (0.74)   |   28.21 (0.29) \| 1.94 (0.30)   |
> | AdaFocal          |    7.56 (0.27) \| 2.69 (0.29)   |    6.79 (0.29) \| 1.44 (0.31)    |   31.27 (0.49) \| 3.42 (0.67)   |   27.89 (0.37) \| 2.75 (0.45)   |
> | CE (baseline)     |    7.14 (0.25) \| 3.86 (0.30)   |    6.85 (0.18) \| 3.10 (0.30)    |   30.36 (0.30) \| 10.0 (0.72)   |   27.15 (0.21) \| 9.09 (0.49)   |
> | SECA (Ours)       |    7.07 (0.20) \| 2.94 (0.35)   |    6.47 (0.22) \| 2.41 (0.37)    |   29.82 (0.22) \| 1.90 (0.61)   |   26.97 (0.11) \| 1.71 (0.47)   |
>
> &nbsp;
>
> > **Weakness 4 - "Missing comparison with recent 2023–2024 calibration methods."**
>
> In the current comparison, we have included several well-known and SoTA calibration work around 2023, for example, DualFocal is from ICML 2023, Adafocal is from NeurIPS 2022, and MDCA is from CVPR 2022. If there are specific 2023-2024 works that reviewer believes are particularly important to compare with, we would be happy to incorporate them into the revision or discuss their relation to SECA.
>
> &nbsp;
>
> > **Weakness 5 - "No analysis of failure modes or dataset characteristics where SECA underperforms."**
>
> Based on the current empirical results across all architectures and datasets evaluated, we didn’t observe obvious failure patterns relative to Cross-Entropy or other methods. However, we observe that 1) SECA’s advantage narrows when batch-size become extremely small (as shown in Appendix D); 2) on datasets that are already intrinsically well-calibrated such as CIFAR-10 where model deep networks exhibit low mis-calibration, the absolute margins between calibration methods is shrink.

---

> ### Author Response · Authors · 2025-11-26
>
> > **Question 1 - "How exactly is μ_j computed when class j has no samples in the current batch?"**
>
> As defined in Eq. 5 and 6, the batch-level average $\mu\_{j}$ is computed only when class $j$ is preset in the batch. If a class doesn’t appear in the current batch, SECA will be reduced the standard Cross-Entropy for that class. No undefined computation happens. Moreover, even when class $j$ is absent, its logits are still updated through samples of other classes, as all logits participate in the softmax normalisation. According to this fact, we designed SECA by leveraging the batch-level class-wise distribution, to provide a stronger and more calibrated corrective signal. For a sample with label $y\_{i}$, the gradient on an absent class $j$ includes the additional term, $\mu\_{y\_{i}, j}$, which suppresses spurious confidence more effectively than standard CE. Thus, decision boundaries involving classes with zero or few batch samples continue to be shaped indirectly. As empirical evidence, our ImageNet results with batch size 1024, and the ablation studies in Appendix D support that SECA remains stable and effective even when per-class samples are sparse.
>
> &nbsp;
>
> > **Question 2 - "What is the minimum batch size required for SECA to work effectively?"**
>
> Empirically, the ablation studies in Appendix D evaluates this question, the results of batch-sizes from 8-128 on CIFAR-10/100 show that SECA remains stable across all tested sizes, and consistently improves calibration over standard Cross-Entropy loss, with larger batches would provide stronger benefits due to richer class-level statistics. In the extreme case such as 1 sample per batch, SECA will reduce to standard Cross-Entropy, but the training remains well-defined and doesn’t adversely affect performance.
>
> &nbsp;
>
> > **Question 3 - "How does SECA perform on severely imbalanced datasets?"**
>
> Although our main experiments do not include artificially imbalanced datasets, SECA’s formulation is inherently compatible with class imbalance. Similar to our discussion in Q1, when minority class samples appear infrequently, SECA still benefits from calibrated suppression of spurious confidence via interactions with majority class samples through the softmax coupling and the $\mu_{j}$ term. For example, our experiments on CIFAR-100 and ImageNet datasets, although both datasets consist of balanced samples per class, mini-batch sampling doesn’t guarantee balanced representation, especially when the number of classes is substantially larger than the batch size.  The strong performance of SECA on these datasets suggests that it handles sparse per-class cases robustly.
>
> We agree that a dedicated long-tailed benchmark would further validate this point, and we will include it in the revised version once the experiments (on the imbalance CIFAR-100 with 10/50/100 imbalance factors) are finished.
>
> &nbsp;
>
> > **Question 4 - "Can you provide confidence intervals and statistical significance tests?"**
>
> Please see our response for Weakness 3.
>
> &nbsp;
>
> > **Question 5 - "How are hyperparameters for baseline methods chosen? Using original paper settings may not be fair."**
>
> For all compared methods, we followed the hyper-parameter setups recommended in their original papers, which is a standard practice adopted in the prior works, e.g., DualFocal, AdaFocal, MDCA, etc. Several of these methods involve non-trivial hyper-parameters, e.g., seven hyper-parameters in AdaFocal, that require their paper-specified values to perform as intended. Besides, conducting a full grid search for each baseline across all models and datasets would require substantial computation, particularly large-scale datasets, and is not commonly done in train-time calibration studies.

---

> ### Author Response · Authors · 2025-12-02
>
> As promised in our earlier response, we conducted additional experiments on imbalanced CIFAR-100 to further examine SECA’s behaviour when minority classes appear infrequently during training. The results and corresponding analysis have included in the revised manuscript as Appendix H. As shown in the table below, the results evaluate SCE under imbalance factors of 10/50/100.
>
>
> |  Methods          | **IF-10** | **IF-50** | **IF-100** |
> |-------------------|-----------|-------------|--------------|
> | Focal      |     3.83    |     5.78    |     6.56     |
> | LS         |     3.53    |     5.89    |     6.92     |
> | MMCE       |     5.40    |     **5.36**    |    **6.19**     |
> | DCA        |     5.10    |     7.63    |     8.48     |
> | FLSD       |     3.74    |     5.89    |     6.80     |
> | MDCA    |     4.27    |     6.51    |     7.00     |
> | Brier             |     6.06    |     9.22    |     9.67     |
> | OLS        |     3.55    |     5.72    |     6.61     |
> | DualFocal |     3.57    |     5.56    |     6.43     |
> | AdaFocal          |     4.15    |     6.43    |     7.38     |
> | CE (baseline)     |     4.47    |     7.15    |     8.25     |
> | SECA (Ours)       |    **3.45**    |     5.46    |     6.42     |
>
>
> Across all three settings, SECA maintains strong and stable calibration performance, it achieves the best SCE under IF-10 and remains highly competitive under IF-50 and IF-100, despite **without extra imbalance-specific tuning**. In contrast, several existing train-time calibration methods, e.g., DCA, AdaFocal, show substantial degradation as imbalance grows. These findings support our claim that SECA’s batch-conditional hybrid target naturally preserves calibration quality even when per-class sample frequencies become highly skewed.
>
> We appreciate your suggestion to evaluate SECA under long-tailed scenarios, and we believe these new results provide clear empirical evidence of SECA’s robustness in such settings.

---

### Official Review · Reviewer_Pt6S · 2025-10-29

**Soundness:** 2
**Presentation:** 3
**Contribution:** 2
**Rating:** 6
**Confidence:** 3

**Summary:**

This paper presents a new training-based calibration method that attempts to prevent overconfidence by using soft targets that combine the one hot ground-truth class with a component based on the batch-averaged prediction for a corresponding class. By doing so, this forgoes the need for additional hyperparameters or significant computational overhead while maintaining competitive performance with alternate calibration training methods across various architectures and datasets. A theoretical analysis is provided as additional motivation for the method along three aspects, showing how soft targets act as a KL-divergence-like regularizer and that it in theory should produce more favourable gradients for balancing out excessive confidence.

**Strengths:**

1. Paper is overall clear and easy to follow. Authors comprehensively test three different kinds of architectures and compare with a wide variety of calibration methods, showing the effectiveness of their method.

2. A great deal of analytical work is presented covering important real-world considerations, such as dataset shift, combination with post-hoc calibration methods, and computational efficiency which helps further motivate the method.

3. The provided theoretical analysis brings further insight and aids to ground the method.

**Weaknesses:**

1. Although the method does appear to be strong and theoretically justified, given the vast numbers of calibration methods already and how temperamental their performance often is depending on various factors, the main advantages of this method are that it is (mostly) parameter free and has a small training footprint, but even in the experimental results for the other methods using the ideal parameters (which is often only 1 parameter) from their original papers without tuning leads to very competitive performance (in fact the performance of many methods is very close to each other). Furthermore, it is stated they were not specifically tuned for each model, so if this was done their performance could be even better, so it is difficult to definitively say SECA is superior with these assumptions.

2. In addition, although the method is hyperparameter free in that it does not introduce new hyperparameters, it is sensitive to batch size which appears to need some optimization based on figures 6-9 in the appendix where in a few cases using higher batch size leads to worse performance (most notably figure 7 on CIFAR-10). This presents a limitation in the method and should be discussed given that it practice it might take optimization for ideal performance.

3. Class-wise calibration consistency is discussed as justification for the method in the Knowledge Distillation Perspective section but no experimental results are provided to validate this assertion. Looking at per-class ECE similar to the MDCA paper would help to validate this point.

4. Table 2 shows computational time for the different methods, but other loss based methods like focal loss that should not add much, if any, noticeable computational overhead are shown to be significantly slower than SECA and cross entropy training which makes it appear that the comparison is not fair despite claiming the same number of training steps. Can the authors comment on this and say how training times were tracked for each calibration method?

**Questions:**

See Weaknesses

---

> ### Author Response · Authors · 2025-11-26
>
> We are grateful for your positive assessment of our work and your thoughtful suggestions, which helped us improve both the empirical analysis and the clarity of the manuscript. We respond to each of your comments in detail below.
>
> > **Weakness 1 - "Given the vast numbers of calibration methods…, the main advantages of this method… but performance of many methods is very close to each other… difficult to say SECA is superior."**
>
> Our intention is not to claim unconditional superiority across all settings, but to demonstrate that SECA can achieve strong and consistent performance under a controlled and commonly adopted training protocol. Specifically, we followed the widely used practice in the prior works, e.g., MDCA, AdaFocal, using each method’s originally recommended hyper-parameters. This protocol ensures comparability and avoids introducing tuning advantages for particular methods. Under this setting, most compared methods indeed perform competitively, but SECA still delivers consistent improvements across different architectural families, and datasets with different modalities (as shown in Tables 1-3). **Importantly, SECA achieves these results without introducing any method-specific hyper-parameters, and its training costs close to Cross-Entropy.** Under a fair and uniform setting, SECA provides a practicable and reliable calibration-performance balance, this is one of major advantages of it.
>
> &nbsp;
>
> > **Weakness 2 - "Although the method is hyperparameter free…, it is sensitive to batch size… needs discussion."**
>
> The ablation studies in Appendix D indicate that SECA consistently outperforms Cross-Entropy in terms of performance and calibration, across entire tested batch sizes on all models and datasets (as shown in Figures 6-9). Although in a few settings, e.g., CIFAR-10, its performance slightly decreases at larger batch sizes, these variations are modest and do not qualitatively change SECA’s advantage.
>
> We agree that extreme batch sizes could affect the calibration strength, so we’ve explicitly discussed this phenomenon and provided a practical guideline in the revised manuscript (highlighted in Appendix D).
>
> *“Overall, these ablation studies show that although SECA naturally inherits a degree of batch-size dependence due to its use of batch-level class-wise statistics, it remains consistently superior to Cross-Entropy across all tested batch sizes and across all datasets and architectures. The modest fluctuations observed in certain configurations, such as larger batch sizes on CIFAR-10, do not change its overall advantage. Importantly, SECA does not require tuning of the batch size to obtain strong results, all main experiments employ standard batch sizes commonly used in prior work, under which SECA remains stable and effective. We acknowledge that extreme batch sizes may affect calibration strength, and we view this as an inherent limitation shared by batch-dependent calibration methods. In practice, SECA provides reliable behaviour under typical training setups.”*
>
> &nbsp;
>
> > **Weakness 3 - "Class-wise calibration consistency is discussed… but no experimental results are provided… looking at per-class ECE would help validate this point."**
>
> While we didn’t include explicit per-class ECE in the submission, our existing analysis provides empirical evidence to support the mechanism discussed in the knowledge distillation perspective.
>
> First, the SCE metric used in the manuscript evaluates calibration on a per-class basis and averages uniformly across all classes, thereby reflecting class-wise calibration behaviour. Second, the anaylsis in Appendix A directly measures class-level alignment through reduced KG divergence and increased cosine similarity between individual predictions, and their corresponding class-level batch average. These results empirically demonstrate that SECA encourages predictions to converge toward class-consistent distributions during training. Besides, Appendix F presents detailed 25-bin reliability diagrams across all datasets, which show calibration improvements are not confined to a subset of classes but occur uniformly across all confidence ranges, thus indicating class-level calibration consistency.

---

> ### Author Response · Authors · 2025-11-26
>
> > **Weakness 4 - "Table 2 shows computational time…, but focal loss… appears significantly slower… comparison not fair… authors should comment."**
>
> We appreciate the reviewer’s attention to the computational cost comparison, and we can confirm that all training times in the Table 2 were measured using the same general training protocol, e.g., hardware, dataloader, number of epochs, etc. The training times were tracked by log files after the training.
>
> The slowdowns for certain methods like focal, MMCE, could be due to: 1) extra per-sample operations inside the loss function, such as exponentiation and normalisation steps over all classes; 2) more expensive gradient computations in each batch. These operations probably won’t introduce noticeable overhead on small datasets like CIFAR, but on large-scale datasets like ImageNet, element-wise exponentiation, kernel operation, etc., could introduce significant accumulated overhead. In contrast, SECA requires only a batch-level class-wise averaging operation, which is inexpensive to compute.

---

### Official Review · Reviewer_JLhG · 2025-10-31

**Soundness:** 2
**Presentation:** 3
**Contribution:** 2
**Rating:** 2
**Confidence:** 4

**Summary:**

This paper proposes an adaptive loss modification approach applied within each training batch. By refining the loss function across all output classes, the method aims to enhance model calibration.

**Strengths:**

1. The experimental section is comprehensive, and the results clearly demonstrate the method’s effectiveness in improving calibration.
2. The paper provides a multifaceted analysis of the method from entropy, gradient, and knowledge-distillation perspectives.

**Weaknesses:**

1. The motivation for this approach remains unclear. Neither the motivation section nor the discussion of loss-function issues sufficiently explains the rationale behind the proposed method.
2. The logical explanation of how the method improves calibration lacks clarity. It is not evident how the proposed approach ultimately enhances model calibration. Moreover, while the authors acknowledge similarities with existing methods, they do not adequately distinguish their approach from prior work.
3. The novelty of the method is not sufficiently good.

**Questions:**

1. What is the ultimate outcome toward which the self-guided mechanism drives the model? Specifically, is this self-guided dynamic loss function stable during optimization, and does it converge to a particular state? What are the convergence properties of this method during optimization?
2. What is the underlying motivation for introducing the self-guided mechanism? If its purpose is merely regularization or soft labeling, why is self-guidance necessary? The approach appears to introduce only an additional term during optimization, as shown in Equation (8), which resembles a dynamic soft-labeling strategy.
3. What motivates Equations (5) and (6)? What is the conceptual significance of incorporating the batch-averaged prediction into the hybrid target formulation?
4 . Why does this method not compromise accuracy?

---

> ### Author Response · Authors · 2025-11-26
>
> We appreciate the time you invested in evaluating our work in depth. Your questions helped us to strengthen the conceptual clarity and technical explanations of SECA, which we address point-by-point below.
>
> > **Weakness 1 -"The motivation for this approach remains unclear. Neither the motivation section nor the discussion of loss-function issues sufficiently explains the rationale behind the proposed method."**
>
> The motivation for SECA is discussed across the Introduction, Related Work, and Methodology sections as follows:
>
> 1) Existing train-time calibration loss functions typically introduce additional hyper-parameters, which require careful tuning for each dataset or model (lines 49-50, lines 136-138).
>
> 2) The post-hoc methods offer no impact on the model’s intrinsic calibration behaviour (lines 44-46, lines 107-109).
>
> 3) In Section 3.1, we clearly explained the root cause of mis-calibration, that the vanilla cross-entropy loss is continuously driving the logit upwards (lines 173-174).
>
> Thus, the above points explicitly motivate the need for a hyper-parameter-free and practically deployable alternative, and why SECA is designed. Further, in Section 3.2 (lines 196-211 ), we described how batch-level class-wise predictions capture the model’s collective belief and motivate the use of these statistics to form adaptive targets.
>
> &nbsp;
>
> > **Weakness 2 - "The logical explanation of how the method improves calibration lacks clarity. It is not evident how the proposed approach ultimately enhances model calibration."**
>
> We provide a step-by-step logical explanation of how SECA improves calibration from three perspectives in Section 3.3 as follows:
>
> 1) The entropy/KL perspective, where Eq. 11 shows that SECA introduces a KL term that penalises overconfident deviations (lines 251-259).
>
> 2) The gradient perspective, where Eq. 12-14 demonstrate that SECA counteracts cross-entopy's persistent upward logit pressure for the target class and aligns with non-target probabilities with class-wise consistent batch behaviour.
>
> 3) The knowledge distillation perspective, where Eq. 15 explains that batch-level class-wise distribution $\\mu_{y\_{i}}$ acts as an adaptive in-batch “teacher”, leading predictions toward stable class-conditional distributions.
>
> We also provide empirical evidence in **Appendix A (page 13-15)**, to support the above theoretical analyses, which compared the training dynamics across SECA, cross-entropy, and label smoothing on CIFAR-10/100 and NLP datasets. The results show consistent trends in entropy, KL divergence, and cosine similarity that align with the theoretical analyses.
>
> &nbsp;
>
> > **Weakness 3 - "Novelty of the method is not sufficiently good."**
>
> Regarding the novelty, we’d like to clarify that SECA differs substantially from existing calibration methods. As discussed in Related Work section, and Section 3.2, prior train-time approaches either rely on fixed hyper-parameters, e.g., label smoothing, focal loss, meta-learned parameters, e.g., AdaFocal, or auxiliary accuracy-based regularisers like MDCA. In contrast, SECA introduces a hyper-parameter-free, per-class, batch-conditional target formulation, which dynamically adapts to the model’s evolving predictive behaviour, without extra hyper-parameter tuning or validation sets. This design yields a novel form of self-guided class-aware target adjustment that is not present in the previous work.  If the reviewer is aware of any specific closely related prior work that we’ve missed, we are happy to clarify the difference and novelty of our work relative to any missed prior work.
>
> &nbsp;
>
> > **Question 1 - "What is the ultimate outcome toward which the self-guided mechanism drives the model? Is this dynamic loss function stable during optimization, and does it converge?"**
>
> SECA’s optimisation outcome is presented and discussed in Section 3.3., and optimisation stability is presented in the Appendix A. In Section 3.3, we discussed SECA from a knowledge-distillation perspective, that the batch-level class-wise distribution $\\mu_{j}$ serves as an adaptive in-batch “teacher”. And Eq. 15 describes a natural fixed point where individual predictions converge toward the class-conditional batch average $\\mu_{j}$. Furthermore, in Appendix A, figures 2-5 empirically demonstrate stable convergence behaviour across all datasets, where KL divergence between predictions and batch-level class-wise averages decreases over training, and cosine similarity steadily increases.

---

> ### Author Response · Authors · 2025-11-26
>
> > **Question 2 - "What is the underlying motivation for introducing the self-guided mechanism? If its purpose is merely regularization or soft labeling, why is self-guidance necessary? The approach appears to introduce only an additional term during optimization, as shown in Equation (8), which resembles a dynamic soft-labeling strategy."**
>
> We’ve partially addressed this in our response on motivation for **Weakness 1**, and mechanism of SECA in **Weakness 2**. Further, as discussed in the Related Work and Section 3.2, SECA differs fundamentally from generic regularisation or soft labelling schemes. Specifically, instead of applying uniform or fixed smoothing, SECA constructs batch-level class-wise hybrid targets using $\\mu_{j}$. This enables adaptive, hyper-parameter-free adjustment that reflects the model’s class-specific predictive behaviour.
>
> &nbsp;
>
> > **Question 3 - "What motivates Equations (5) and (6)? What is the conceptual significance of incorporating the batch-averaged prediction into the hybrid target formulation?"**
>
> As discussed in section 3.2, the batch-level class-wise distribution $\\mu_{j}$ captures the model’s collective belief over samples belonging to the same class in the current batch. This distribution serves as a class-conditional estimate of the model’s current predictive tendency, allowing SECA to form hybrid targets that adapt to class-specific calibration patterns, and provide a principled and data-driven way to adjust target sharpness based on model’s evolving behaviour. The empirical studies on the training dynamics in Appendix A can support our claim.
>
> &nbsp;
>
> > **Question 4 - "Why does this method not compromise accuracy?"**
>
> As presented in the experiment section, SECA doesn't only compromise accuracy but also often improves it. The formulation of SECA inherently preserves accuracy as the hybrid target keeps the correct-class component strictly anchored at the one-hot label, which means the primary discriminative gradient which driving the model to make correct prediction remains like cross-entropy. SECA only modifies the confidence sharpness via the additional $\\mu_{j} $ term, it adjusts over-confidence without reducing the gradient magnitude needed to learn class boundaries. For instance, as shown in Eq. 12, the direction for the correct class remains negative as in CE, ensuring that learning of the correct decision boundaries is unaffected.

---

> > ### Comment · Reviewer_JLhG · 2025-11-28
> > **Response to Rebuttal**
> >
> > Thank you for your response. Although you have provided analysis from three additional perspectives, I still cannot identify the most direct motivation for the proposed method based on the current manuscript. The method incorporates the average predicted probability of each class across batches into the loss function, yet the purpose of this design remains unclear to me. Compared with standard optimization objectives—such as the basic cross-entropy loss—it is not evident what advantage or new direction this modification introduces.
> >
> > Furthermore, while the authors claim that the approach introduces no additional hyperparameters, equations (6) and (8) still apply a weighting term, even if its coefficient is fixed at 1.
> >
> > Most importantly, the manuscript does not clearly articulate the fundamental motivation behind this approach. As shown in equation (8), the formulation simply adds a penalty term to reduce the probability assigned to a particular class—a strategy that is not novel and has been widely used for a long time.

---

> ### Author Response · Authors · 2025-11-28
>
> Thanks for your follow-up. We now provide a more direct statement of SECA’s core design intention and how the batch-averaged term serves that intention. Below is a more focused clarification:
>
> > **1. Direct motivation for SECA and the role of batch-averaged term**
>
> As discussed in the manuscript, our goal is to introduce a self-adaptive, class-aware calibration signal that counteracts the persistent logit-inflation behaviour of standard Cross-Entropy, without adding dataset- or model-dependent hyper-parameters. Standard Cross-Entropy always pushes the logit of ground-truth class upward, even then the prediction is already very confident, and it never explicitly encourages class-wise calibration consistency (lines 152-166).
>
> The batch-level distribution $\\mu\_{j}$ is introduced precisely to address the aforementioned issue, it captures the current collective  behaviour of the model for class $j$ in that batch, Using $\\mu\_{y\_{i}}$ in Eq. 6 and Eq. 8 ensures that:
>
> 1) the additional term always reflects what the model currently 'believes' for that class, rather than a fixed prior or manually chosen target;
> 2) the strength and direction of the calibration pressure automatically adapts per class and per batch, instead of being controlled by an explicit scalar hyper-parameter.
>
> This is the **most direct** motivation for incorporating the batch average, as it provides a data-driven, class-conditional teacher that continuously adjusts calibration strength as training progresses.
>
> &nbsp;
>
> > **2. Advantage and direction relative to basic Cross-Entropy**
>
> As shown in Eq. 11, SECA can be written as:
>
> $$
> \\mathcal{L}\_{\\mathrm{SECA}} = - \\log p\_{i,y\_{i}} + KL(\\boldsymbol{\\mu}\_{y\_{i}} \| \\mathbf{p}\_{i}) + H(\\boldsymbol{\\mu}\_{y\_{i}}),
> $$
>
> so relative to vanilla Cross-Entropy, SECA adds a class-conditional KL regulariser that drives each prediction $p\_{i}$ toward the batch-informed distribution $\\mu\_{y\_{i}}$. This differs from standard objectives in two important ways:
>
> 1) The regulariser is not a generic "confidence penalty" but a dynamic KL term whose target is the current batch-level class distribution. When the model is over-confident for a class, $\\mu\_{y\_{i}}$ becomes sharper and the KL term penalises deviations accordingly, when predictions are diffuse, the same term encourages concentration, as shown expirically in Appendix A.
> 2) The regulariser is per-class and per-batch, so different classes naturally receive different calibration strengths depending on their current behaviour, which we found to be crucial on datasets like CIFAR-100 and ImageNet where class-wise behaviour is highly heterogeneous.
>
> Empirically, this design yields substantial improvements over basic Cross-Entropy in calibration metrics across datasets with different modalities, while preserving or improving accuracy (as shown in Tables 1-3).
>
> &nbsp;
>
> > 3. **"No additional hyperparameters" and the fixed weighting term**
>
> The Eq. 6 and Eq. 8 may looks like applying a weighting term, by "no additional hyper-parameters" we specifically mean that:
>
> 1) SECA doesn’t introduce any tunable scalars, e.g., $\\alpha$, $\\beta$, $\\gamma$, whose value must be selected per dataset or architecture;
> 2) the coefficient of the KL term is fixed as part of the definition of SECA, and is not tuned or adjusted during experiments.
>
> The usage is consistent with how "hyper-parameter-free" is typically used in the literature. For instance, methods like Label Smoothing, Focal Loss or AdaFocal, etc., all include at least one scalar that must to tuned or meta-learned for good performance, while SECA doesn’t require such a calibration-strength knob.
>
> &nbsp;
>
> > 4. **Novelty beyond "adding a penalty to reduce probability of a class"**
>
> Our contribution is specific as: SECA uses a batch-conditioned, per-class teacher $\\mu\_{j}$ rather than a fixed prior, uniform target, or hand-crafted confidence penalty. To our knowledge, prior calibration methods either 1) rely on fixed soft labels; 2) introduce tunable focusing/smoothing parameters; or 3) add auxilary accuracy-based terms, e.g., MDCA. They do not combine a hyper-parameter-free, batch-level, class-conditional teacher with the specific gradient structure we derive and empirically validate. If the reviewer is aware of any prior work which is specifically close to SECA, please inform us, we are happy to clarify the difference and novelty of our work relative to it.

---

### Official Review · Reviewer_2j5J · 2025-10-31

**Soundness:** 2
**Presentation:** 3
**Contribution:** 2
**Rating:** 4
**Confidence:** 3

**Summary:**

This paper proposes a new train-time calibration method called SECA (Self-guided Model Calibration). The method dynamically adjusts the training target distribution by fusing the batch-averaged model predictions with the ground-truth labels, thereby achieving self-adaptive confidence calibration. The authors argue that this approach can improve the alignment between predictive confidence and true accuracy without introducing additional hyperparameters. The paper further provides theoretical analysis from the perspectives of entropy regularization, gradient dynamics, and knowledge distillation. Extensive experiments across different model architectures (e.g. CNN, ViT, BERT) and tasks in vision and language domains show that SECA achieves better calibration performance compared with traditional cross-entropy and other existing calibration methods.

**Strengths:**

1. The paper is well-structured with a clear logical flow and appropriate level of detail.
2. The theoretical analysis is logically coherent and easy to follow.
3. The experiments are diverse and conducted across multiple architectures/datasets.

**Weaknesses:**

1. The form of $\hat{q}$ should be more precise, as there may be a normalization process that has been omitted.
2. The derivation of Equation (12) is not accurate. The derivative of the first term should be $p_{i,c} - q_{i,c}$, but the derivative of the latter term may not simply be $-\mu_{y,c}$; thus, the equation may not hold as written.
3. The experiments are not comprehensive and consequently not trustworthy. In comparison papers, results are often reported both before and after grid search temperature tuning, but this paper does not include such comparisons.

**Questions:**

Why were ResNet-32 and ResNet-56 chosen, while previous works like DualFocal[1] and AdaFocal[2] typically used ResNet-50 and ResNet-110?

Other questions are listed in the Weaknesses section.

[1] Linwei Tao, Minjing Dong, and Chang Xu. Dual focal loss for calibration. In International Conference on Machine Learning, pp. 33833–33849. PMLR, 2023.
[2] Arindam Ghosh, Thomas Schaaf, and Matthew Gormley. Adafocal: Calibration-aware adaptive focal loss. Advances in Neural Information Processing Systems, 35:1583–1595, 2022.

---

> ### Author Response · Authors · 2025-11-26
>
> We sincerely thank you for your careful and constructive review, and for the detailed technical feedback that helped us identify and clarify several important aspects of the paper. Below, we address them point-by-point.
>
> > **Weakness 1 - "The form of $\tilde{q}$ should be more precise, as there may be a normalization process that has been omitted."**
>
> We thank the reviewer for pointing this out regarding the hybrid target $\tilde{q}$, it’s defined as:
> $$
> \\tilde{q}_{i,c} = q\_{i,c}\+\\mu\_{y_i,c},~\\text{where}\~
> q\_{i,c} =
> \\begin{cases}
> 1, & \\text{if } c = y_i, \\\\
> 0, & \\text{otherwise}.
> \\end{cases}
> $$
>
> where $q\_{i}$ is the one-hot ground-truth vector (sums to 1) and $\\mu\_{y_i}$ is the class-wise batch-averaged prediction (also sums to 1). Thus, $\\sum\_{c=1}^C{\\tilde{q}\_{i,c}}=2$. Using an unnormalised hybrid target is an intentional design choice of SECA, i.e., keeping both components at their natural scale allows SECA to amplify the corrective signal derived from $\\mu\_{y_i}$, strengthening the calibration effect.
>
> To avoid ambiguity, we have altered the following statement after the Eq. 6 in the revised manuscript:
> “Note that, the hybrid target $\\tilde{\\mathbf{q}}\_{i}$ is intentionally unnormalised. This preserves the full influence of both the one-hot label and the class-conditional batch-averaged prediction. Cross-Entropy with unnormalised non-negative targets is mathematically valid, and the resulting gradient naturally matches the refined expression in Eq. 12.”
>
> &nbsp;
>
> > **Weakness 2 - "The derivation of Equation (12) is not accurate. The derivative of the first term should be … but the derivative of the latter term may not simply be …; thus, the equation may not hold as written."**
>
> We acknowledge that the earlier form of Eq. 12 implicitly assumed a normalised target distribution, which was inconsistent with the intentionally unnormalised hybrid target defined in Eq. 6 (as we explained in above weakness point). We have corrected this oversight by refining the gradients using the correct form of $\tilde{q}_{i}$. Since the elements of $q\_{i} $ and $\mu\_{y_i}$ each sum to 1, the hybrid target sums to 2, this leads to refined Eq. 12 as:
>
> $$\\frac{\\partial \\mathcal{L}\_{i}}{\\partial z\_{i,c}} = 2p\_{i,c} - (q\_{i,c} + \\mu\_{y\_{i},c})$$
>
> Accordingly, we have also updated Eq. 12-14 and rewritten accompanying descriptions in the “Gradient Perspective” paragraph, to explain the impact of this amplified gradient and its role in SECA’s calibration behaviour.
>
> &nbsp;
>
> > **Weakness 3 - "The experiments are not comprehensive and consequently not trustworthy. In comparison papers, results are often reported both before and after grid search temperature tuning, but this paper does not include such comparisons."**
>
> Our initial submission included these results, specifically, **the pre- and post- temperature-scaled performance for all methods on CIFAR-10 and CIFAR-100 in Table 5 (Appendix E)**, and also referenced these results in the main content (lines 469-470). As shown there, SECA maintains consistent calibration improvements in both evaluation settings.
>
> Besides, we have also included empirical studies (in Appendix A) to support our theoretical analysis (discussed in Section 3.3), ablation studies to investigate the impacts of varying batch sizes (in Appendix D), comprehensive reliability analysis (in Appendix F), and OoD detection experiments (in Appendix G).
>
> &nbsp;
>
> > **Question 1 - "Why were ResNet-32 and ResNet-56 chosen, while previous works like DualFocal and AdaFocal typically used ResNet-50 and ResNet-110?"**
>
> We followed the exact experimental setup used in MDCA [1], which also adopts ResNet-32/56 as its primary CNN backbones on CIFAR. For large-scale experiments on ImageNet, we adopted Vision Transformer (ViT) as a representative of large vision models. ViT represents a modern and widely used architectural family distinct from CNNs, and including it broadens the architectural diversity of our evaluation. This choice is better at revealing the generalisability of calibration methods across different model families rather than restricting the analysis to CNN variants alone.
>
>  &nbsp;
>
>  [1] Hebbalaguppe et al., A stitch in time saves nine: A train-time regularizing loss for improved neural network calibration. CVPR 2022.

---

### Author Response · Authors · 2025-11-26

We thank all reviewers for their great efforts and constructive comments. We also appreciate AC and SAC for overseeing the review process. Below, we summarise some of the key responses and updates below:

**1) Clarification of the hybrid target and gradient formulation**

We clarified that the hybrid target in SECA is intentionally unnormalised and corrected Eq. 12 accordingly. The revised gradients now fully align with actual formulation. These updates resolve the concerns from **Reviewer 2j5J** and ensure mathematical consistency throughout Section 3.3.

**2) Motivation and mechanism**

We clarified the motivation across the Introduction, Related Work, and Section 3.1, emphasising the need for hyper-parameter-free train-time calibration method. The entropy/KL, gradient, and knowledge-distillation perspectives collectively explain how SECA improves calibration, and Appendix A provides empirical validation. These responses address the **Reviewer JLhG**’s concerns.

**3) Breadth and fairness of experiments**

The manuscript included pre-/post- temperature-scaling results, batch-size ablation studies, training dynamics studies, reliability diagrams, and OoD robustness experiments, in the Appendix part.  These empirical results provide a comprehensive evaluation across CNN, ViT, and BERT. Architectural choices follow SoTA works like MDCA for CIFAR, and we use ViT to broaden model diversity. All methods were trained under the same general training setup, e.g., batch size, number of epochs, etc., for method-specific hyper-parameters, we use recommended setups in their original papers.

**4) Behaviour under small or incomplete batches**

We clarified that SECA doesn’t assume every class appears in each batch. When a class is absent, the loss reduses to CE for that class while logits continue to update through softmax coupling. Ablation studies in Appendix D show stable performance for batch sizes from 8-128, compared to CE. This resolves concerns from **Reviewers Pt6S and 1vsK**.

---

### Meta-Review · Area_Chair_toSe · 2026-01-06

**Summary:**

1. The novelty and fundamental motivation of the proposed SECA method were heavily questioned by reviewers. Reviewer JLhG consistently argued that the method lacked a clear, novel motivation beyond adding a batch-conditioned penalty, viewing it as a non-novel extension of existing soft-labeling techniques. The authors specifically responded to Reviewer JLhG's critique on the novelty and motivation of their method. However, their response appears to have failed to satisfy the reviewer's fundamental question.

2. Rreviewers Pt6S pointed out that while SECA performs well, its performance is often very close to other, carefully-tuned baseline methods. The claimed benefits of being hyperparameter-free and efficient were deemed insufficiently groundbreaking.

3. Concerns about theoretical depth and guarantees were raised by reviewers. The multi-perspective analysis (entropy, gradient, distillation) was seen by some as a post-hoc explanation rather than a driving theoretical advance, lacking rigorous characterization of when the method succeeds or fails.

**Reviewer Concerns:**

**Concerns Addressed by the Rebuttal:**

*Regarding the presentation (pointed by reviewer 2j5J).*

The authors corrected the equations and clarified the intentional use of an unnormalized hybrid target, resolving concerns about gradient derivation and normalization.

*How does SECA perform on severely imbalanced datasets?*

The authors added new experiments on imbalanced CIFAR-100 to demonstrate robustness across varying batch sizes and class distributions.

*The comparison regarding computational cost is not fair.*

The authors explained the runtime differences, attributing them to intrinsic per-sample operations in other loss functions.

**Still Outstanding Concerns:**

The novelty and motivation are the most significant unresolved issues. The rebuttal did not change the core critique that SECA's mechanism is perceived as an incremental combination of existing ideas (batch statistics + soft labels) rather than a novel conceptual idea. The motivation for why this specific fusion constitutes a significant advance remains unclear.

The concern that SECA's performance is very competitive but not decisively superior to other methods, and that its "hyperparameter-free" nature is a modest incremental benefit, was not fully addressed.

**Reviewer Scores:**

It appears that reviewer JLhG who initially gave a score of 2 was not at all persuaded by the authors' rebuttal, and I believe this reviewer would not change their score. The other three reviewers have not yet engaged in further discussion with the authors, and their motivation for potentially revising their scores remains unclear.

---

### Decision · Program_Chairs · 2026-01-26

Reject